

# From models to SMEFT and back?

Ilaria Brivio[1], Sebastian Bruggisser[1], Emma Geoffray[1*], Wolfgang Kilian[2],
Michael Krämer[3], Michel Luchmann[1], Tilman Plehn[1] and Benjamin Summ[3,4]

**1** Institut für Theoretische Physik, Universität Heidelberg, Germany
**2** Department of Physics, University of Siegen, Germany
**3** Institut für Theoretische Teilchenphysik und Kosmologie,
RWTH Aachen University, Germany
**4** Institut für Theoretische Physik und Astrophysik, Universität Würzburg, Germany

⋆ geoffray@thphys.uni-heidelberg.de

## Abstract

We present a global analysis of the Higgs and electroweak sector, in the SMEFT framework and matched to a UV-completion. As the UV-model we use the triplet extension of the electroweak gauge sector. The matching is performed at one loop, employing functional methods. In the SFitter analysis, we pay particular attention to theory uncertainties arising from the matching. Our results highlight the complementarity between SMEFT and model-specific analyses.

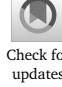

## Content

# 1   Introduction

The Higgs discovery [1,2] and many measurements of the Higgs Lagrangian [3] indicate that the Standard Model with its single, weakly interacting Higgs boson might well be the correct effective theory around the electroweak scale. However, the Standard Model is extremely unlikely to be the full story. Many theoretical considerations, including electroweak baryogenesis, dark matter, or neutrino mass generation, point to an extended electroweak or scalar sector. To avoid a bias through a specific, pre-selected signal hypothesis, modern LHC searches for beyond the Standard Model (BSM) physics are often conducted in the Standard Model effective theory (SMEFT) [4]. Because of its vast operator landscape, the corresponding experimental searches [5,6] and global analyses [7–13] provide a comprehensive probe of rates and kinematic patterns in LHC processes.

One of the complications of SMEFT analyses of LHC data is that the effective theory truncated at dimension six has a limited validity range, and that LHC measurements span a large energy range. Moreover, even if we assume the SMEFT to be generally valid, it is not clear how much information on a full BSM model is lost when we confront it with LHC data via a truncated SMEFT Lagrangian rather than the original full model. Combining these questions, it is instructive to consider concrete, albeit simplified, BSM models and examine the limits extracted through a SMEFT interpretation matched to these models in comparison with the constraints obtained from direct searches [12,14–21].

The naive expectation behind SMEFT analyses is that we can use the complete, correlated information on the Wilson coefficients from a global analysis and derive limits on any BSM model through matching. However, if the BSM scale is not sufficiently well-separated from the electroweak (EW) scale, an interpretation based on the SMEFT Lagrangian truncated at dimension six will likely give inaccurate results [22,23]. The theory uncertainties related to the matching to full models are usually not accounted for in global analyses, which instead take their Lagrangian as a fixed interpretation framework. In general, limits derived on BSM models through a SMEFT framework using the same data and with all uncertainties accounted for will differ from limits derived on the full model directly, where the former can be significantly weaker or stronger than the latter.

This work aims at exploring the complementarity of the two analysis strategies and at highlighting general aspects that emerge when the SMEFT results are related to a concrete BSM scenario. We address this question for a global analysis of electroweak, di-boson and Higgs measurements, matching the relevant Wilson coefficients to the UV-model at one loop, using functional matching methods. We use the SFITTER framework and include a proper estimate of a new and non-negligible theory uncertainty from the variation of the matching scale. As a UV-model we use a triplet-extended gauge sector [14,24–27] a standard scenario when it comes to motivating the SMEFT approach to the Higgs and electroweak sector. Such a triplet model can be linked for instance to the weakly coupled gauge group $SU(3) \times SU(2) \times SU(2) \times U(1)$ [28] or deconstructed extra dimensions [29].

The paper is organized as follows: in Sec. 2 we review the basics of functional one-loop matching, we define the gauge triplet model under study, and we provide details about the

SFITTER setup. In Sec. 3 we discuss the decoupling limit of the new heavy states and the relevance of the matching scale choice. The impact of these two aspects on the global analysis is illustrated via simplified fits. In Sec. 4 we present the results of a global fit to the full vector triplet model, based on the dimension-6 SMEFT Lagrangian, and compare our results with limits obtained from direct searches. We conclude in Sec. 5.

## 2 Basics

In this section, we briefly review the one-loop matching procedure, the UV-model, as well as the SFITTER setup. Experienced readers are welcome to skip this section.

### 2.1 One-loop matching: generic approach

The methods of constructing and matching effective-field theories [30,31] have been in use for more than four decades [32–35]. Generic expressions for the low-energy effective action of a gauge theory at the one-loop order were derived in the 80s [36]. More recently, the approach has been further explored, particularly within the context of SMEFT [37–50].

We consider a UV model which can be defined in terms of light fields $\psi$ and heavy fields $\Psi$, and which supports a perturbative expansion based on a local Lagrangian. Heavy fields are characterized by the condition that the support of their spectral functions vanishes below a certain threshold. We may identify the threshold with a mass $M$, typically the lightest mass that belongs to the heavy spectrum. The remaining fields are understood as light fields.

The UV model is expressible in terms of an effective action $\Gamma_{UV}[\psi, \Psi]$, the generating functional of its one-particle irreducible (1PI) vertex functions. If fields of spin higher than $1/2$ are involved, or if global symmetries are present, it is constrained to be a solution of a Slavnov-Taylor identity. By assumption, $\Gamma_{UV}$ is calculable in a loop expansion from a local Lagrangian $\mathcal{L}_{UV}(\psi(x), \Psi(x))$ with a finite number of fields and parameters. The parameters depend on the choice of a regularization and renormalization scheme and are redefined order by order by suitable renormalization conditions. This includes resolving inherent ambiguities associated with field reparameterizations, such as wave-function renormalization and terms vanishing by equations of motion.

The EFT is likewise expressible in terms of an effective action $\Gamma_{EFT}[\psi]$, a functional of the light fields only. Again, we assume that a perturbative loop expansion is possible, and that it can be computed from a local Lagrangian $\mathcal{L}_{EFT}(\psi(x))$. The number of parameters of $\mathcal{L}_{EFT}$ is intended to be finite, but it increases without bounds with the accuracy that we want to implement via matching conditions. To keep the EFT parameter set manageable, we have to define an organizing principle which amounts to a series of approximations, and a prescription to truncate this series at a certain order.

To find the EFT Lagrangian iteratively, one introduces the one-light-particle irreducible (1LPI) effective action $\Gamma_{L,UV}[\psi]$. Formally, this is a double Legendre transform of $\Gamma_{UV}[\psi, \Psi]$; in practice, it amounts to absorbing a maximal set of independent heavy-field propagators in the skeleton expansion of S-matrix elements. This results in redefined light-field effective vertices. By contrast, the light-field propagators are kept explicit. In general they still carry a mixture of light and residual heavy degrees of freedom, depending on the precise definition of the original UV model. Like the original effective action, $\Gamma_{L,UV}[\psi]$ depends on conventions regarding renormalization and handling the equations of motion. In terms of this entity, the matching condition reads

$$\Gamma_{L,UV}[\psi] = \Gamma_{EFT}[\psi] + \Delta\Gamma[\psi].\tag{1}$$

The matching error $\Delta\Gamma[\psi]$ describes a set of vertex-function corrections $\Delta\Gamma_i(x)$ that are not calculable from a local Lagrangian involving light fields only. The matching procedure succeeds if, in momentum space, all contributions to this error are sufficiently power-suppressed at low energy,

$$\Delta\Gamma_i(p) < c|p|^k, \tag{2}$$

where $p$ is any light-particle mass or momentum component.

At the tree level, the 1LPI effective action $\Gamma_{\text{L,UV}}^{(0)}[\psi]$ of the UV model can be derived by simple variable changes, applying the equations of motion. Unless the $\psi$ multiplets are incomplete under a symmetry, the result satisfies the tree-level Slavnov-Taylor identity with only light fields taken into account. The tree-level effective action $S_{\text{EFT}}[\psi] = \Gamma_{\text{EFT}}^{(0)}[\psi]$ is evaluated, to arbitrary order, by means of a momentum-space Taylor expansion of the 1LPI effective action on the l.h.s. of Eq.(1). In this expansion, residual heavy degrees of freedom are naturally removed from the tree-level light-field propagators. The latter assume their canonical tree-level form while any extra terms are shifted to the interaction part of $S_{\text{EFT}}[\psi]$.

The operator content of the tree-level effective action $S_{\text{EFT}}[\psi]$ can be determined independently by algebraic methods. Their coefficients are fixed by a term-by-term comparison with the vertices of $\Gamma_{\text{L,UV}}^{(0)}$. The symmetries are preserved in this expansion if covariant derivatives are used consistently. At one loop, new contributions to the UV effective action arise which are generically non-local, and can be formally summarized as

$$\Gamma_{\text{UV}}^{1\ell}[\psi, \Psi] = ic_s \operatorname{Tr} \log\left(-\frac{\delta^2 S_{\text{UV}}[\psi, \Psi]}{\delta^2(\psi, \Psi)}\right), \tag{3}$$

where the trace is integrated over all field components at all space-time points and $c_s$ accounts for the statistics of the fields that are integrated over. This evaluates to the sum of all one-loop Feynman graphs with external fields attached. In expressions of this kind, the external field insertions act as bookkeeping devices, or background fields [51–59]. This allows for employing gauges and conventions that distinguish between internal and external lines, a generic feature of working with 1PI vertex functions. This means in particular that, with respect to the background fields, a manifestly gauge-invariant effective action can be computed [60,61]. The trace is in general UV divergent and requires the application of a regularization scheme and the addition of local counterterms, such as dimensional regularization and minimal subtraction.

To match the UV model to the EFT at the one-loop order, we have to evaluate Eq.(1) again. Initially,

$$\Delta\Gamma^{1\ell}[\psi] = ic_s \operatorname{Tr}\left[\log\left(-\frac{\delta^2 S_{\text{UV}}[\psi, \Psi]}{\delta^2(\psi, \Psi)}\right) - \log\left(-\frac{\delta^2 S_{\text{EFT}}^{(0)}[\psi]}{\delta^2 \psi}\right)\right]\Bigg|_{\Psi=0}, \tag{4}$$

where the formal trace includes the integral over all one-loop diagrams which are 1LPI and do not contain open external $\Psi$ lines. Because $S_{\text{EFT}}^{(0)} = \Gamma_{\text{EFT}}^{(0)} = \Gamma_{\text{L,UV}}^{(0)} + \mathcal{O}(|p|^k)$, the difference is well-behaved in the IR. Loops of canonical light propagators only would exactly cancel between the two terms, but since the light-field propagators need not coincide between the two Lagrangians, we have to be careful to take all terms into account. In any case, due to the IR cancellation the one-loop functional Eq.(4) again admits a Taylor expansion up to the order of the previous tree-level truncation. The result can be expressed as a finite set of local terms that modify the coefficients of terms which are already present in the generic effective Lagrangian of the tree-level EFT. They are absorbed in $S_{\text{EFT}}[\psi]$,

$$S_{\text{EFT}}^{1\ell}[\psi] = -\Delta\Gamma^{1\ell}[\psi]\Big|_{\text{local, truncated}}, \tag{5}$$

and disappear from Eq.(4). In effect, the remainder still contains all non-local parts of the matching error but satisfies Eq.(2), to one-loop order.

By the same reasoning, the difference in Eq.(4) is not well-behaved but divergent in the UV, and therefore requires regularization and renormalization. The renormalization conditions are given by the matching conditions themselves and thus indirectly refer to the renormalization conditions of the UV model. All free parameters of the EFT are fixed, order by order, in terms of the original parameters of the UV model. Nevertheless, a practical scheme such as dimensional regularization with minimal subtraction may introduce an intermediate renormalization which depends on an arbitrary scale $\mu_R$. The implications of this additional mass scale will be discussed in detail below.

In analogy with the tree-level matching procedure, in order to manifestly preserve the symmetries of the theory one should consistently work with covariant derivatives in the one-loop matching calculation, as discussed in the following subsection. However, due to the presence of UV divergences in the matching conditions the Slavnov-Taylor identity need not be compatible with a local Taylor expansion of the one-loop vertex functions, and the separation of the UV effective action into a gauge-invariant low-energy effective action and a remainder like in Eq.(1) may fail [36, 62, 63]. In the current paper, we assume that such an obstruction does not critically affect our argument.

## 2.2 One-loop matching: implementation

Instead of constructing the difference in Eq.(4) in terms of Feynman graphs explicitly, the subtraction may be accounted for in the integrand by employing the method of regions [64–66]. The matching correction (Eqs.(4) and (5)) is replaced by

$$S^{1\ell}_{\text{EFT}}[\psi] = i c_s \, \text{Tr} \log \left( -\frac{\delta^2 S_{\text{UV}}[\Psi, \psi]}{\delta(\Psi, \psi)^2} \right) \Bigg|_{\text{hard}} . \tag{6}$$

The label 'hard' has to be understood in the following way: the functional trace is computed in momentum space. Two different regions are of interest in the matching, the hard and the soft region. If $q$ denotes the typical size of a loop momentum, the hard region is defined by $q \sim M \gg m$, whereas the soft region is defined by $q \sim m \ll M$. Here $m$ stands for the typical mass scale of the light sector. As discussed above, only the hard region is relevant while in the soft region the matching integral is well behaved. It has been shown that the tree-level induced EFT contribution to the matching cancels the soft region contributions from the UV-theory in the difference in Eq.(4) [42, 67]. Therefore, the integrands of the loop integrals in Eq.(6) are expanded only in the hard region. The evaluation of the functional trace then reduces to computing integrals of the form

$$\int \frac{\mathrm{d}^d q}{(2\pi)^d} \frac{q^{\mu_1} \dots q^{\mu_{2n_c}}}{(q^2 - M^2_{i_1})^{n_1} \dots (q^2 - M^2_{i_m})^{n_m} (q^2)^{n_0}} . \tag{7}$$

Here, all masses $M_{i_1}, \dots, M_{i_m}$ are of the order of $M$. This implies that the dependence of Eq.(6) on any external momentum or mass $|p|$ is analytic, and no logarithms of the form $\log(m/|p|)$ or $\log(|p|/M)$ can appear. The only logarithm possible is $\log(M/\mu_R)$, and to avoid large logarithms in the relation between EFT and UV parameters we need to choose $\mu_R \sim M$.

Apart from the prescription 'hard', the second derivatives of the UV-action evaluated at the background field configurations appear in the matching. To derive a universal result these derivatives are split into a part that contains the gauge-kinetic term of the field and its mass term, generating the propagator of the field, and a pure interaction contribution that appears

in the final result. For the field $\psi$ this latter piece is given by

$$X_{\psi\psi} = -\frac{\delta^2 S_{\text{UV,int.}}}{\delta\psi^2}, \tag{8}$$

where only the interaction part of the action excluding the interactions with gauge bosons through the covariant derivative appears. The interactions with the gauge bosons are included in the propagator part of the functional derivative, which allows for an evaluation in which only gauge covariant objects appear at every step and the final result is manifestly gauge invariant. The price to be paid for this manifest gauge covariance is that every occurrence of a covariant derivative has to be shifted by a loop momentum in the evaluation of the functional trace in Eq.(6). We therefore have to parameterize Eq.(8) as

$$X_{\psi\psi} = U_{\psi\psi} + iD_\mu Z^\mu_{\psi\psi} + iZ^{\dagger\mu}_{\psi\psi}D_\mu + \dots, \tag{9}$$

where $D_\mu$ is the covariant derivative of the UV-model. The quantities $U_{\psi\psi}, Z^\mu_{\psi\psi}$ and $Z^{\dagger\mu}_{\psi\psi}$ only depend on covariant derivatives through commutators whereas the explicit covariant derivatives appearing in Eq.(9) are so-called open covariant derivatives that act on everything to their right. The ellipsis denotes terms with further open covariant derivatives. Importantly, contributions with one open covariant derivative arise at dimension six whenever there is a scalar field charged under the gauge group and therefore they contribute to the matching through the presence of the Higgs field. Consequently, for our matching computations we use an extension of the results of Ref. [44], adding gauge bosons and the heavy resonance of our model. Since the gauge boson fluctuations appear in loops they have to be gauge fixed. This gauge fixing does not disturb the manifest gauge invariance at the level of the background fields and the gauge-fixing parameter can be chosen at convenience. Choosing Feynman gauge allows for easy incorporation of these operators into the results of Ref. [44], since we can treat gauge bosons like scalar fields with an extra index. Care has to be taken to account for the overall sign in the propagator. For the resonance this choice is not available since it does not have a gauge-fixing term and some operators with up to two open covariant derivatives have to be computed for the matching.

## 2.3 Triplet model

The UV model we study in this paper is a gauge-triplet extension of the Standard Model [14, 24–27]. In the unbroken electroweak phase, the Lagrangian reads

$$\mathcal{L} = \mathcal{L}_{\text{SM}} - \frac{1}{4}\widetilde{V}^{\mu\nu A}\widetilde{V}^A_{\mu\nu} - \frac{\tilde{g}_M}{2}\,\widetilde{V}^{\mu\nu A}\widetilde{W}^A_{\mu\nu} + \frac{\tilde{m}^2_V}{2}\widetilde{V}^{\mu A}\widetilde{V}^A_\mu$$
$$+ \sum_f \tilde{g}_f\,\widetilde{V}^{\mu A}J^{fA}_\mu + \tilde{g}_H\,\widetilde{V}^{\mu A}J^{HA}_\mu + \frac{\tilde{g}_{VH}}{2}\,|\phi|^2\widetilde{V}^{\mu A}\widetilde{V}^A_\mu, \tag{10}$$

where $\widetilde{V}^A_\mu$ is a new, massive vector field transforming as a triplet of $SU(2)_L$, $\widetilde{W}^A_\mu$ are the SM weak gauge bosons, and $\phi$ is the SM Higgs doublet. The kinetic term of the vector field includes a covariant derivative,

$$\widetilde{V}^A_{\mu\nu} = \widetilde{D}_\mu\widetilde{V}^A_\nu - \widetilde{D}_\nu\widetilde{V}^A_\mu \quad \text{with} \quad \widetilde{D}_\mu\widetilde{V}^A_\nu = \partial_\mu\widetilde{V}^A_\nu - g_2 f^{ABC}\widetilde{W}^B_\mu\widetilde{V}^C_\nu, \tag{11}$$

where $A, B, C$ are $SU(2)_L$ indices and the covariant derivative carries a tilde to indicate that it contains the fields $\widetilde{W}^A_\mu$. The currents coupling the heavy vector to the SM-fields are given by

$$J^{lA}_\mu = \bar{l}_i\gamma_\mu t^A l_j\,\delta^{ij}, \qquad J^{qA}_\mu = \bar{q}_i\gamma_\mu t^A q_j\,\delta^{ij}, \qquad J^{HA}_\mu = \phi^\dagger i\overleftrightarrow{D}^A_\mu\phi, \tag{12}$$

with $l, q$ being the SM lepton and quark doublets, $t^A = \sigma^A/2$ the $SU(2)$ generators and $\sigma^A$ the Pauli matrices. $i, j$ are flavor indices and the Lagrangian is defined in a flavor-symmetric limit. In the Higgs current, $(\phi^\dagger i \overleftrightarrow{D}^A_\mu \phi) = i\phi^\dagger t^A(D_\mu\phi) - i(D_\mu\phi^\dagger)t^A\phi$. As pointed out in [67], the theory cannot be quantized in a self-consistent way for $\tilde{g}_{VH} < 0$.

The gauge mixing described by the triplet model is familiar from the general case of extra-$U(1)$ bosons [68]. A special feature is the explicit $\widetilde{V}$-mass term, which would have to be generated by some kind of symmetry breaking and likely involve additional fields; we ignore these additional fields, for instance in their effect on $\tilde{g}_M$. The Higgs doublet $\phi$ is yet to develop a VEV, which means we are working in the unbroken electroweak phase. Underlying this choice is the assumption that a SMEFT expansion for the EFT exists. This is the case unless there are additional sources of electroweak symmetry breaking, or a heavy particle obtains all of its mass from the Higgs VEV [69]. Even in the weakly coupled UV-completion of the triplet model there are no additional sources of electroweak symmetry breaking, because the additional scalar breaks $SU(2) \times SU(2)$ to $SU(2)_L$ and leaves the electroweak symmetry completely intact.

To remove the kinetic mixing, we can re-define the SM-gauge field as [25, 26]

$$W^{\mu\nu A} = \widetilde{W}^{\mu\nu A} + \tilde{g}_M \widetilde{V}^{\mu\nu A} = \partial^\mu(\widetilde{W}^{\nu A} + \tilde{g}_M \widetilde{V}^{\nu A}) - \partial^\nu(\widetilde{W}^{\mu A} + \tilde{g}_M \widetilde{V}^{\mu A}) + \cdots. \tag{13}$$

For the triplet field we only allow for a re-scaling, $\widetilde{V}^{\mu A} = \alpha V^{\mu A}$, so that the triplet mass does not get transferred into the SM-gauge sector. The triplet mass also fixes the phase of the real vector field $\widetilde{V}^A_\mu$, such that $\alpha$ has to be real. Requiring a canonical normalization of the new kinetic term $V^{\mu\nu A}V^A_{\mu\nu}$, we find $\alpha^2 = 1/(1 - \tilde{g}_M^2)$. This relation requires $\tilde{g}_M \neq \pm 1$, to ensure a valid model with a propagating heavy vector. Furthermore, as we will see in Sec. 3, we need to require $|\tilde{g}_M| < 1$ for the squared pole mass of the resonance to be positive. The final form of the gauge field re-definition in Eq.(13) becomes

$$\widetilde{W}^{\mu A} = W^{\mu A} - \frac{\tilde{g}_M}{\sqrt{1 - \tilde{g}_M^2}} V^{\mu A} \qquad \text{and} \qquad \widetilde{V}^{\mu A} = \frac{1}{\sqrt{1 - \tilde{g}_M^2}} V^{\mu A}, \tag{14}$$

and brings the Lagrangian into the form

$$\begin{aligned}
\mathcal{L} = \mathcal{L}_{\text{SM}} &- \frac{1}{4} V^{\mu\nu A}V^A_{\mu\nu} + \frac{m_V^2}{2} V^{\mu A}V^A_\mu \\
&+ \sum_f g_f V^{\mu A}J^{fA}_\mu + g_H V^{\mu A}J^{HA}_\mu + \frac{g_{VH}}{2} |H|^2 V^{\mu A}V^A_\mu \\
&+ \frac{g_{3V}}{2} f^{ABC} V^{\mu A}V^{\nu B}V^C_{\mu\nu} - \frac{g_{2VW}}{2} f^{ABC} V^{\mu B}V^{\nu C}W^A_{\mu\nu}, \tag{15}
\end{aligned}$$

which has the same structure as Eq.(10), but additional triple and quartic gauge couplings between the weak and triplet sectors. The Lagrangian parameters are related through

$$m_V^2 = \frac{\tilde{m}_V^2}{1 - \tilde{g}_M^2}, \qquad\qquad g_H = \frac{\tilde{g}_H + g_2\tilde{g}_M}{\sqrt{1 - \tilde{g}_M^2}}, \qquad\qquad g_f = \frac{\tilde{g}_f + g_2\tilde{g}_M}{\sqrt{1 - \tilde{g}_M^2}},$$

$$g_{VH} = \frac{2\tilde{g}_{VH} + g_2^2\tilde{g}_M^2 + 2g_2\tilde{g}_H\tilde{g}_M}{2(1 - \tilde{g}_M^2)}, \quad g_{3V} = -\frac{2g_2\tilde{g}_M}{(1 - \tilde{g}_M^2)^{1/2}}, \quad g_{2VW} = \frac{g_2\tilde{g}_M^2}{1 - \tilde{g}_M^2}, \tag{16}$$

where $g_2$ denotes the $SU(2)_L$ gauge coupling. The heavy vector triplet couples to the weak gauge bosons not only via the $g$−couplings in Eq.(15), but also through the non-abelian component of the covariant derivative Eq.(11), that leads to interaction terms of the form $(\partial V)VW$ and $VVWW$. These interactions are weighted by the weak gauge coupling, and therefore are present even if $g_i(\tilde{g}_i) \equiv 0$.

## 2.4 SFitter setup

The SFITTER framework [70] has been long employed for global analyses of LHC measurements in the context of Higgs couplings and EFTs [7,71–75], including a comprehensive study of an analysis in terms of Higgs couplings and its UV-completion [76]. The approach is unique in that it allows a comprehensive treatment of uncertainties: SFITTER uses a likelihood set up that includes a broad set of statistical, systematic, and theory uncertainties. Statistical and most systematic ones are described by a Poisson- or Gauss-shaped likelihood. Theoretical uncertainties lack a frequentist interpretation, and are described by flat likelihoods in SFITTER, corresponding to a range of equally likely theory predictions. An important difference between employing a flat likelihood compared to a Gaussian one is that the uncorrelated profile likelihood adds the uncertainties from the flat distributions linearly, while Gaussian error bars are added in quadrature. The profile likelihood combination of a flat and a Gaussian uncertainty gives the well-known RFit prescription [77]. Correlations among certain classes of systematic uncertainties are also included.

From the technical point of view, the new aspect of the SFITTER analysis presented in this paper is the translation of the SMEFT likelihood into the parameter space of the UV model. In the fit, all observables are parameterized in the SMEFT using the operator set provided in Tab. 1, that is based on the HISZ basis [78]. All SMEFT predictions are at LO in QCD and scaled by the same corrections as the SM-rates used for the actual experimental analysis. Terms obtained from squaring amplitudes with one operator insertion, that are quadratic in the Wilson coefficients, are retained. The Wilson coefficients are then expressed in terms of $\tilde{g}_i$ parameters of the UV model, Eq.(10), using the one-loop matching expressions onto the Warsaw basis provided in Ref. [79] and the Warsaw-to-HISZ basis translation in Appendix A.1. In this way, the likelihood can be directly sampled in the parameter space of the UV model.

In addition, we employ a new likelihood sampling method [80] compared to previous SFITTER analyses, that ensures a much more efficient sampling close to the SM point, where all Wilson coefficients vanish. By contrast, the previous sampling method was optimized for the detection of potential secondary maxima in the likelihood, by giving higher weight to the edges of the parameter space.

### Dataset

The SMEFT analysis presented in this work builds directly on the dataset employed in Ref. [7], which includes electroweak precision observables (EWPO) at LEP (14 measurements), Higgs measurements (275) and di-boson measurements at the LHC (43). The latter contain results from both Run 1 and Run 2 [74]. In addition, we include differential measurements from three resonance searches by ATLAS, that reach up to invariant masses in the multi-TeV range and that we re-interpret within the SMEFT framework. One of these [81] was already included in the analysis of Ref. [7]. The other two [82, 83] are more recent and have been added specifically for this work. These measurements are not usually included in the SMEFT analyses and are not covered by the simplified template cross section framework [84]. Nevertheless, it can be instructive to explore their sensitivity, particularly to operators that induce momentum-enhanced corrections. Moreover, all the resonance searches considered here target heavy vector triplets decaying into $WH$ or $WW$ as a potential signal. Therefore they allow to compare directly the constraining power of the SMEFT analysis to that of the direct search.

### Theory uncertainties

In view of the upcoming LHC runs and their rapidly growing data sets, the treatment of theory uncertainties in global analyses is becoming critical. In our analysis, we include theory uncertainties associated to parton distribution functions, to missing higher orders in the SM or

SMEFT predictions, and to the matching scale to the EFT. The latter will be discussed in more detail in Sec. 3.2.

For the time being, we do not include uncertainties associated to missing SMEFT operators due to the truncation of the SMEFT Lagrangian [85] or to symmetry assumptions, such as CP-conservation. Nevertheless, the impact of missing higher orders in the EFT expansion becomes obviously manifest in the comparison between constraints extracted from the SMEFT analysis and from direct searches.

Concerning higher orders in the loop expansion, Higgs analyses in SFITTER currently adopt the most accurate SM predictions available, which are implemented so as to match the state-of-the art predictions reported in the experimental analyses. The corresponding $K$-factors are then applied onto the tree-level SMEFT predictions as well, which is tantamount to assuming that QCD corrections scale evenly for all SMEFT operators and in the same way as in the SM. Although this assumption is, strictly speaking, not correct [86], for the rate measurements considered here we do not expect large variations in the $K$-factors between different operators. For some kinematic distributions these effects can be larger. We therefore assign conservative theory uncertainties in order to reduce the numerical impact of these effects. A proper SMEFT simulation of Higgs and di-boson production up to NLO in QCD is postponed to a future work.

## 3 Toy fits and matching uncertainty

In this section we discuss two aspects of the vector triplet model and of its matching onto the SMEFT, that are preliminary to a correct SMEFT global analysis. The first is the decoupling limit of the model, and the second is the numerical impact of varying the scale at which the 1-loop matching is performed. Both issues are analyzed via simplified toy fits.

### 3.1 Decoupling

The decoupling limit of the vector triplet model considered in this work is most easily identified starting from the Lagrangian of Eq.(15), where, as long as the EW symmetry is unbroken, the heavy triplet and the SM gauge bosons do not mix. In this case, it is easy to see that the BSM states decouple for large values of the physical mass, $m_V \to \infty$. This is directly reflected in the matching formulas, which give $\lim_{m_V \to \infty} C_i \equiv 0$ for all the dimension-6 Wilson coefficients. At the level of a global fit, the decoupling limit can be visualized by setting the central values of all the measurements included to match the corresponding SM predictions. Figure 1 shows the results obtained in this way from SFITTER: the likelihood is first computed as a function of 4 free parameters in the Lagrangian of Eq.(10)

$$\{\tilde{m}_V, \tilde{g}_M, \tilde{g}_H, \tilde{g}_l\}, \tag{17}$$

setting other parameters $\tilde{g}_q$, $\tilde{g}_{VH}$ to zero. We then project them onto the 7 parameters for the rotated Lagrangian of Eq.(15),

$$\{m_V, g_H, g_l, g_q, g_{VH}, g_{3V}, g_{2VW}\}. \tag{18}$$

At this stage, we fix the matching scale to $Q = m_V = 4\,\text{TeV}$. For each of the couplings we see that, as expected, the range of allowed values increases as $m_V^{-1} \to 0$. It is worth noting that the rate at which this happens varies between the $g$-parameters. This is due to the fact that the matching expressions do not scale homogeneously with $g_i^2/m_V^2$, but generally have a more complex polynomial structure. The degeneracy between $g_i$ and $1/m_V$ in these expressions is also broken by the $V - W$ interactions proportional to the weak gauge coupling. The homogeneity of the yellow regions indicates that there the likelihood is flat and no point is

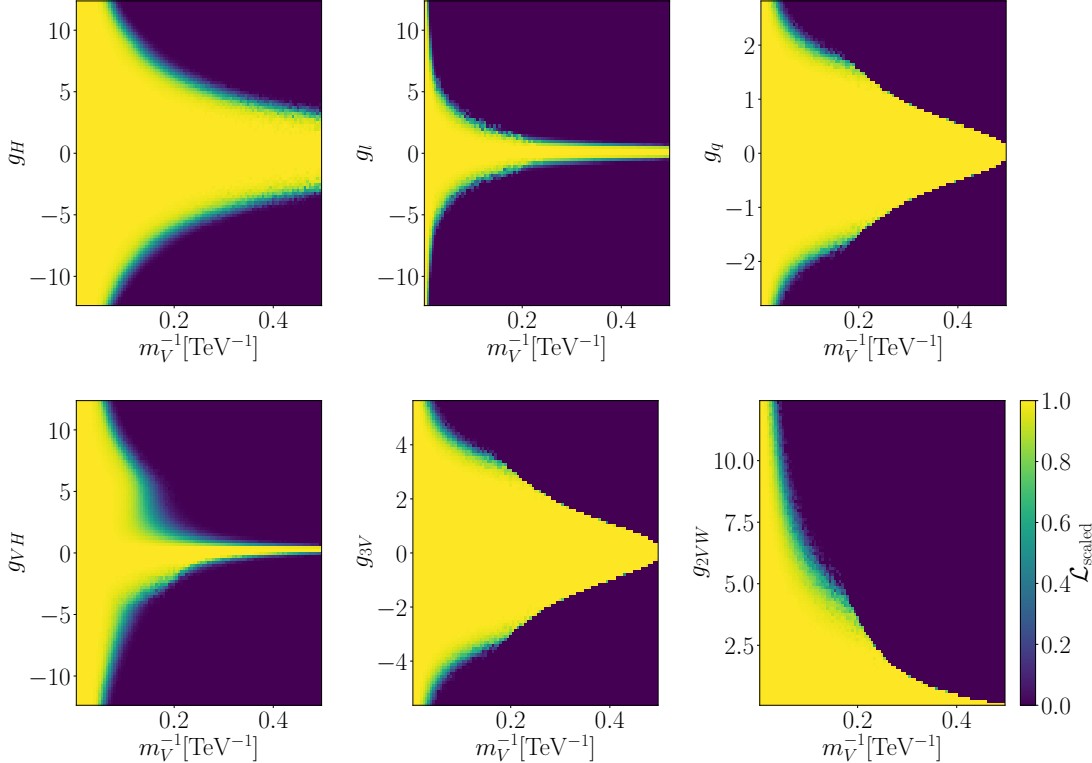

Figure 1: Decoupling pattern for the vector triplet model. Global fit with all measurements at their SM values and to the 4 free parameters $\tilde{m}_V, \tilde{g}_M, \tilde{g}_H, \tilde{g}_l$, and subsequently projected onto the 7 parameters of the unmixed Lagrangian Eq.(15).

preferred. Setting all measurements to their actual measured values, which generally depart from the SM predictions, has the effect of introducing a substructure in the likelihood, thereby identifying a more restricted preferred region. This is shown, for a subset of panels, in Fig. 2. Here, for instance, the best fit point moves to finite $m_V$ and prefers non-vanishing values of $g_H$. Note that, to good approximation, the entire region highlighted in green is allowed at 68%CL. The yellow points simply identify a best-fit region and should not be interpreted as statistically significant. Finally, the reduced number of parameters in the Lagrangian Eq.(10) as compared to the setup without kinetic mixing induces strong correlations through $\tilde{g}_M$, as illustrated in the $g_{2VW} - g_{3V}$ plane of Fig 2.

As the matching procedure highlighted in Sec. 2.1 requires a separation between light and heavy degrees of freedom, defining the decoupling limit in the notation of Eq.(10) requires some more care, due to the explicit kinetic mixing between the heavy triplet and the SM gauge fields.

From Eq.(16), we see that $m_V \to \infty$ can be achieved for $\tilde{m}_V \to \infty$ or for $|\tilde{g}_M| \to 1$. However, the condition $|\tilde{g}_M| = 1$ does not lead to a well-defined decoupling condition, because in this limit $\tilde{V}_\mu^A$ become auxiliary fields, i.e. the theory loses three dynamical degrees of freedom. This is not sufficient for a proper decoupling in the EFT sense because even as an auxiliary field $\widetilde{V}_\mu^A$ still induces mass-suppressed vertices that enter correlation functions and we enter a strongly interacting regime where our perturbative approach fails.

To see the impact of $\tilde{g}_M$ we resum insertions of gauge mixing into the $\widetilde{W}_\mu^A$ and $\widetilde{V}_\mu^A$ propa-

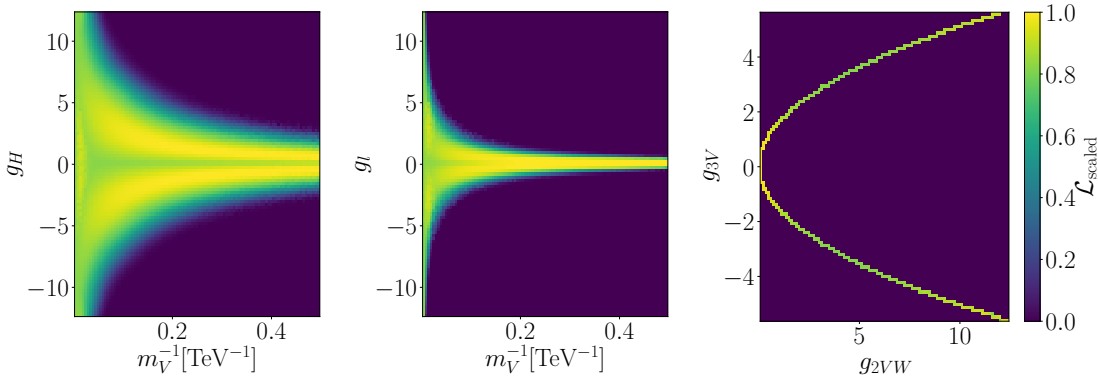

Figure 2: Results of the same global analysis as in Fig. 1, but with measurements set to their actual values.

gators. The corrected propagators of these fields become

$$
\hat{D}_{\mu\nu}^{\widetilde{V}} = -\frac{i}{p^2 - \tilde{m}_V^2 - \tilde{g}_M^2 p^2}\left(g_{\mu\nu} - (1-\tilde{g}_M^2)\frac{p_\mu p_\nu}{\tilde{m}_V^2}\right),
$$

$$
\hat{D}_{\mu\nu}^{\widetilde{W}} = -\frac{i}{p^2}\left(g_{\mu\nu} - (1-\xi)\frac{p_\mu p_\nu}{p^2}\right) - \frac{i\tilde{g}_M^2}{p^2 - \tilde{m}_V^2 - \tilde{g}_M^2 p^2}\left(g_{\mu\nu} - \frac{p_\mu p_\nu}{p^2}\right). \tag{19}
$$

It is easy to see that for $|\tilde{g}_M| = 1$ the resummed $\widetilde{V}_\mu^A$ propagator loses its momentum dependence, which is indicative of the field becoming auxiliary. For $|\tilde{g}_M| > 1$, $\widetilde{V}_\mu^A$ becomes tachyonic while, for $|\tilde{g}_M| < 1$, $\widetilde{V}_\mu^A$ is a dynamical degree of freedom. In this case its propagator has a physical pole at $p^2 = m_V^2$ as defined in Eq.(16), and it can be expanded in $p^2/m_V^2 \ll 1$. The resummed $\widetilde{W}_\mu^A$ propagator includes a term with a pole at $p^2 = m_V^2$, contaminating $\widetilde{W}_\mu^A$ with a contribution from $\widetilde{V}_\mu^A$. Therefore this field cannot be directly identified with the SM weak bosons. However, in the tree-level matching procedure, once the 1LPI effective action is expanded in $p^2/m_V^2$, the component associated with the $\widetilde{V}_\mu^A$ pole is shifted from the propagators to the interaction terms, which are unambiguously fixed at this order by the matching condition of Eq.(2). At one loop, the fact that the EFT is the low-energy limit of the UV model is manifest in the fact that only the 'hard' region of the momentum integral contributes to the functional trace in the matching formula of Eq.(6). As a consequence, the first term of the $\widetilde{W}_\mu^A$ propagator cancels against the corresponding EFT contributions, while the second term genuinely contributes to the matching in the hard region. Equivalently, one can match in the shifted basis directly identifying $W_\mu^A$ in the UV model with the corresponding weak bosons in the SMEFT.

In the top (bottom) panels of Fig. 3 we again show the results of a global analysis where all measurements are set to their SM prediction (to their actual values), this time projected onto a subset of the $\tilde{g}$-parameters and onto the combination $\tilde{g}_M/\sqrt{1-\tilde{g}_M^2}$ that drives most $\tilde{g}-g$ relations, see Eq.(16). For reference, the right panels also show lines of constant $m_V$, such that the decoupling limit $m_V \to \infty$ flows orthogonally to the lines. Consistent with the results in the unmixed basis (Fig. 1), the expected likelihood is mostly flat in the entire preferred region, while the observed one exhibits a substructure that identifies a best-fit region where $\tilde{g}_H \neq 0$ and both $m_V$ and $\tilde{m}_V$ are finite. The reason can be identified in a few EWPO measurements that exhibit small ($< 1\sigma$) deviations from the SM expectation: $A_l(SLD)$ and $m_W$.

For $|\tilde{g}_M| \to 1$ the theory becomes strongly interacting and some perturbative unitarity considerations are therefore pertinent. Requiring the couplings of the unmixed UV theory to

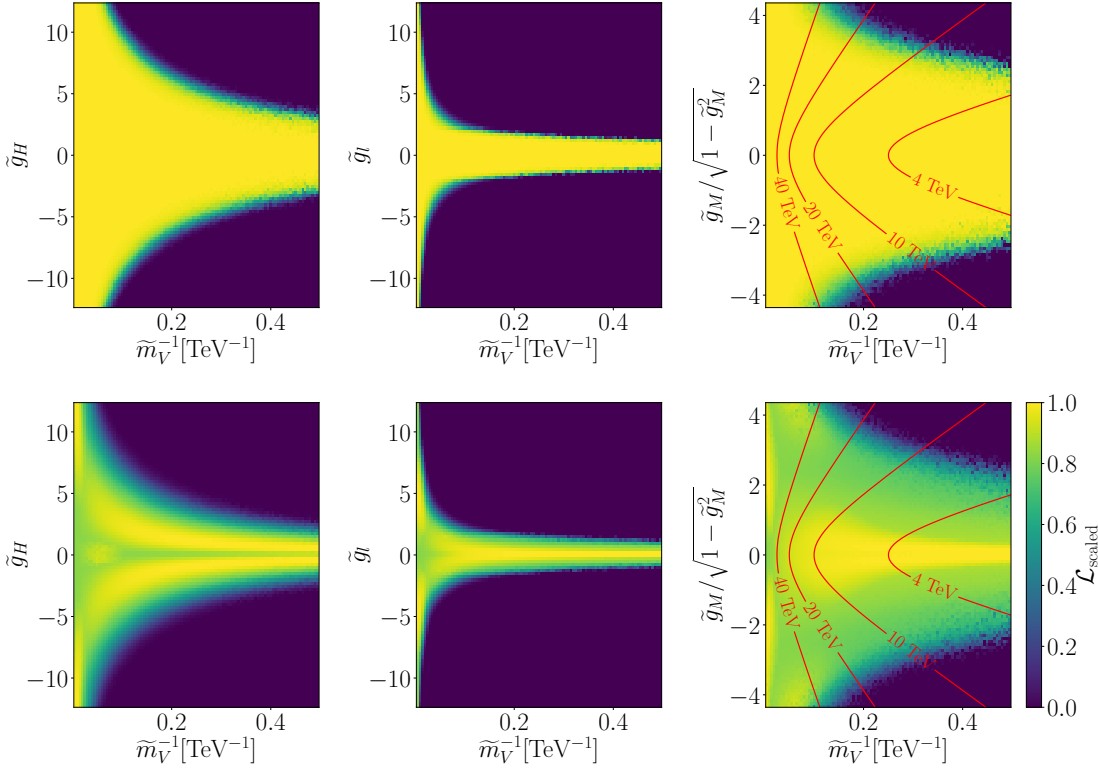

Figure 3: Results of the same global analyses as in Figs. 1 (upper) and 2 (lower), projected on $\tilde{m}_V, \tilde{g}_H, \tilde{g}_l$ and the combination $\tilde{g}_M/\sqrt{1-\tilde{g}_M^2}$.

remain perturbative, the most stringent constraints on $\tilde{g}_M$ stem from $g_{2VW}$

$$g_{2VW} \approx \frac{g_2 \tilde{g}_M^2}{1-\tilde{g}_M^2} < 4\pi \qquad \Leftrightarrow \qquad |\tilde{g}_M| < 0.975\,. \qquad (20)$$

Therefore for all our fits we require $|\tilde{g}_M| < 0.975$.

## 3.2 Matching scale

In perturbative predictions of LHC observables, at least two unphysical scales are known to reflect a theory uncertainty, the factorization scale and the renormalization scale. Both arise from a separation of an observable into different regimes with different perturbative expansions, and the scale dependence would vanish if we would include all orders in all predictions. For a calculation at finite perturbative order we instead use the scale variation as one measure of a theory uncertainty and treat it as an unphysical nuisance parameter in theory predictions [75, 87].

One unphysical scale is the renormalization scale, which in the context of dimensional regularization appears as a free parameter. In more physical terms, the renormalization scale is the energy scale associated with those observables that we select for defining the numerical parameters of the theory, the renormalization conditions. Whenever scale choices are arbitrary, we often identify them with each other and a typical energy scale of the scattering process to avoid large logarithms. Clearly, this does not work if renormalization conditions involve widely distinct energy scales, such as in the relation of UV-model parameters to the low-energy observables of the SM.

The renormalization group equation apparently solves this problem. It relates observables at different scales, properly resumming logarithms and absorbing them into running parameters. However, it works only in the absence of mass thresholds. This strongly suggests to match a UV model with a heavy mass $M$ to a low-energy EFT even if the algebraic simplifications of the latter are not essential for a specific calculation.

In a one-loop matching calculation that uses dimensional regularization, the matching scale enters as an additional parameter. However, in contrast to the original renormalization scale this parameter is not entirely arbitrary. If we want to avoid large logarithms, its reasonable range is bounded from above and below. In line with the generic discussion of one-loop matching above, we illustrate this property in the following section. We consider examples of increasing complexity, starting from the QCD coupling, turning next to the SM extended by a scalar singlet and finally returning to the vector triplet model of Sec. 2.3.

**Running strong coupling**

We can illustrate the appearance of the matching scale using the simple example of the running strong coupling. It provides the key ingredients to understanding the EFT matching scale: the separation of low-energy and UV regimes and contributions beyond tree level. In general, the relation between the bare coupling and the renormalized coupling in the $\overline{\text{MS}}$ scheme is

$$\alpha_s^{\text{bare}} = \alpha_s(p^2)\left[1 - \alpha_s b_0\left(\frac{1}{\bar{\epsilon}} + \log\frac{\mu_R^2}{p^2}\right)\right], \quad \text{with} \quad b_0^{(n_f)} = \frac{1}{4\pi}\left(\frac{11}{3}N_c - \frac{2}{3}n_f\right). \tag{21}$$

Here, $p^2$ is the energy scale of the scattering, $\mu_R^2$ is introduced by dimensional regularization, and $1/\bar{\epsilon} = 1/\epsilon - \gamma_E + \log 4\pi$. We identify our UV-regime as momenta above the top mass, with six propagating quark flavors, and the low-energy regime as described by five propagating quark flavors. The running of $\alpha_s$ in the two regimes is described by the beta function with five or six flavors, respectively. The UV-divergences in the low-energy and full UV-theories arise from five or six propagating flavors, so the renormalization prescription Eq.(21) is different in the two regimes.

The low-energy and UV-regimes are separated by a matching scale $Q$, which we choose to be of the order of the top mass to avoid large logarithms or inconsistent symmetry structures. Matching conditions guarantee that the two predictions for any observable are the same at least at this scale. Instead of looking at a full set of amplitudes or correlation functions, we limit ourselves to the quasi-observable $\alpha_s$. Following Eq.(21), the definitions of $\alpha_s(p^2)$ in relation to the bare parameter are different, but they have to agree when evaluated at the matching scale. This defines a threshold correction

$$1 - \frac{\alpha_s b_0^{(6)}}{4\pi}\left(\frac{1}{\bar{\epsilon}} + \log\frac{\mu_R^2}{p^2}\right)\Bigg|_{Q^2} = 1 - \frac{\alpha_s b_0^{(5)}}{4\pi}\left(\frac{1}{\bar{\epsilon}} + \log\frac{\mu_R^2}{p^2}\right)\Bigg|_{Q^2} + \frac{\alpha_s}{6\pi}\log\frac{\mu_R^2}{Q^2}. \tag{22}$$

The relation of the threshold correction to loop effects is reflected in the logarithmic form $\log\mu_R^2/Q^2$. Together with the five-flavor $\overline{\text{MS}}$ counter term it defines $\alpha_s$ in the low-energy regime as

$$\alpha_s^{\text{bare}} = \alpha_s(p^2)\left[1 - \frac{\alpha_s b_0^{(5)}}{4\pi}\left(\frac{1}{\bar{\epsilon}} + \log\frac{\mu_R^2}{p^2}\right) + \frac{\alpha_s}{6\pi}\log\frac{\mu_R^2}{Q^2}\right]. \tag{23}$$

This definition includes three scales for a given scattering process, the physical scale $p^2$, the renormalization scale $\mu_R^2$, and the matching scale $Q^2$. In simple problems, the renormalization scale and the physical scale can be identified to avoid potentially large logarithms. The matching scale is usually set to the mass of the decoupled particle, $Q = m_t^2$, leading to a threshold correction that is non-zero in general.

From our toy example we can immediately see the role of the threshold correction at the matching scale and the renormalization group running. If we start from the UV, all parameters of the theory evolve based on the full particle spectrum. In the low-energy theory part of the spectrum decouples also from the running, which can even break the underlying symmetries [88], and we will follow a completely different renormalization group flow. The matching corrections adjust for this effect. They move us to the same flow line in the EFT, independent of the choice of matching scale and with all the caveats of maintaining perturbative control, accounting for changes of the spectrum, changing symmetries, etc.

**Singlet extension**

When we interpret a SMEFT calculation for an LHC process as a low-energy approximation to a UV-prediction, we again break the phase space of the scattering process into two parts. We first illustrate SMEFT matching using the singlet-extended SM [89, 90],

$$\mathcal{L} \supset \frac{1}{2}\left(\partial_\mu S\right)\left(\partial^\mu S\right) - \frac{1}{2}M^2 S^2 - A|\phi|^2 S - \frac{\kappa}{2}|\phi|^2 S^2 - \frac{\mu}{3!}S^3 - \frac{\lambda_S}{4!}S^4. \tag{24}$$

The singlet mass is given by $M_S^2 = M^2 + \mathcal{O}(v^2)$; we integrate it out under the condition $M_S \sim M \gg v$, ensuring a consistent expansion in $v/M$ [69]. As a simplification, we also assume $A$ to be of the order of $M$. The leading term in $v/M$ is defined by $v = 0$ and can be obtained by matching in the unbroken phase. In the broken phase the Higgs VEV enters via the masses of the SM-particles which properly belong to the EFT Lagrangian, below the matching scale. Matching in the broken phase would allow us to include partial higher-order corrections in the EFT expansion [23]. Since the mass scales in question are not widely separated, it depends on the detailed numerics which setup yields a more reliable approximation. The SMEFT Lagrangian reads

$$\mathcal{L}_{\text{SMEFT}} = \mathcal{L}_{\text{SM}} + \sum_i f_i(p/\mu_R)\mathcal{O}_i, \tag{25}$$

where the Wilson coefficients are scale dependent. Specifically, we want to define these coefficients such that the SMEFT reproduces all low-energy observables of the UV-theory up to $\mathcal{O}(v^3/M_S^3)$. As matching condition we use Eq.(1). In the functional approach we compute this once and for all using functional traces. To illustrate some features related to the matching scale, we compute some contributions to the Wilson coefficient $f_{\phi,2}$ of the operator $\mathcal{O}_{\phi,2} = \partial_\mu(\phi^\dagger\phi)\partial^\mu(\phi^\dagger\phi)/2$ diagrammatically. As discussed in Appendix A.1, it is related to $Q_{\phi\square} = |\phi|^2 \square |\phi|^2$ as $c_{\phi\square} \approx -f_{\phi,2}/2$, modulo fermionic operators. The operator contributes to the correlation function with two external fields $\phi$ and two external fields $\phi^\dagger$ and depends on $p^2$, so we fix it by requiring

$$\left.\partial_{p^2}\Gamma_{\text{SMEFT}}(\phi^\dagger,\phi^\dagger,\phi,\phi)\right|_{p^2=0} = \left.\partial_{p^2}\Gamma_{\text{L,UV}}(\phi^\dagger,\phi^\dagger,\phi,\phi)\right|_{p^2=0}, \tag{26}$$

order by order in the coupling. With some abuse of notation we also denote specific correlation functions by $\Gamma$, arguments indicating the external fields. Since both sides of the equation involve running parameters, the matching has to be imposed at a given scale,

$$\partial_{p^2}\left(\begin{array}{c}\rule{0pt}{1.5em}\end{array}\otimes\begin{array}{c}\rule{0pt}{1.5em}\end{array} + \text{SM}\right) = \partial_{p^2}\left(\begin{array}{c}\underset{S}{\rule{0pt}{1.5em}}\end{array} + t\text{-channel} + \text{SM}\right) \quad \text{at } p^2 = 0.$$

The SM-contributions contain the same diagrams on both sides, with appropriately adjusted parameters through the matching conditions, so their contributions cancel. Only diagrams

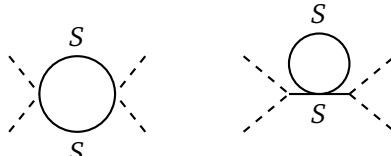

Figure 4: Feynman diagrams contributing to $f_{\phi,2}^{(1)}$. Left: Diagram yielding a $\kappa^2$-contribution. Right: Diagram yielding a $A^2\lambda_S/M^2$-contribution. The dashed line corresponds to the Higgs field, whereas the solid line corresponds to the singlet.

with at least one heavy propagator actually contribute to the matching, so Eq.(26) becomes

$$\partial_{p^2}\left(8p^2 f_{\phi,2}^{(0)}\right)\bigg|_{p^2=0} = \partial_{p^2}\frac{2A^2}{4p^2-M^2}\bigg|_{p^2=0} \qquad \Rightarrow \qquad f_{\phi,2}^{(0)} = \frac{A^2}{M^4}. \tag{27}$$

At tree level, the scale dependence only appears implicitly for $A$ and for $f_{\phi,2}^{(0)}$.

Next, we compute the $\kappa^2$-contribution to $f_{\phi,2}^{(1)}$ at one loop. This contribution is induced by the diagram on the left in Figure 4, where the external particles are as specified in Eq.(26). We again set all external scales to $p^2$ and find for the diagram

$$\kappa^2\mu_R^{4-d}\int\frac{d^dq}{(2\pi)^d}\frac{1}{((2p+q)^2-M^2)(q^2-M^2)} = \kappa^2\frac{i}{16\pi^2}B_0(4p^2,M,M),$$

$$\text{with}\quad B_0(4p^2,M,M) = \frac{1}{\bar{\epsilon}} - \log\frac{M^2}{\mu_R^2} + \frac{2p^2}{3M^2} + \mathcal{O}\left(\frac{p^4}{M^4}\right). \tag{28}$$

In the full expression the renormalization scale appears, but taking the derivative in the matching condition for this contribution to $f_{\phi,2}$ removes it,

$$\partial_{p^2}B_0(4p^2,M,M)\bigg|_{p^2=0} = \frac{2}{3M^2} \qquad \Rightarrow \qquad f_{\phi,2}^{(1)} \supset \frac{1}{16\pi^2}\frac{\kappa^2}{12M^2}. \tag{29}$$

Just as at tree level, the matching scale does not appear explicitly.

Finally, we compute the $A^2\lambda_S/M^2$-contribution to $f_{\phi,2}^{(1)}$ to illustrate the appearance of matching scale logarithms. This contribution arises from the diagram on the right in Figure 4. The diagram is not 1PI, but is 1LPI and therefore has to be included in the matching. With all external scales again set to $p^2$ this diagram gives

$$-\frac{\lambda_S A^2}{(4p^2-M^2)^2}\mu_R^{4-d}\int\frac{d^dq}{(2\pi)^d}\frac{1}{q^2-M^2} = -\frac{\lambda_S A^2}{16\pi^2}\frac{M^2}{(4p^2-M^2)^2}\left(\frac{1}{\bar{\epsilon}}+1-\log\frac{M^2}{\mu_R^2}\right). \tag{30}$$

Taking the derivative with respect to $p^2$ and evaluating it at $p^2=0$ we find the one-loop matching condition

$$f_{\phi,2}^{(1)} \supset -\frac{1}{16\pi^2}\frac{\lambda_S A^2}{M^4}\left(-1+\log\frac{M^2}{Q^2}\right), \tag{31}$$

where the Wilson coefficient explicitly depends on the matching scale. This scale dependence is expected since the corresponding correlation function is divergent. As mentioned before, in models with one new mass scale, we can of course avoid these logarithms by identifying $Q = M$.

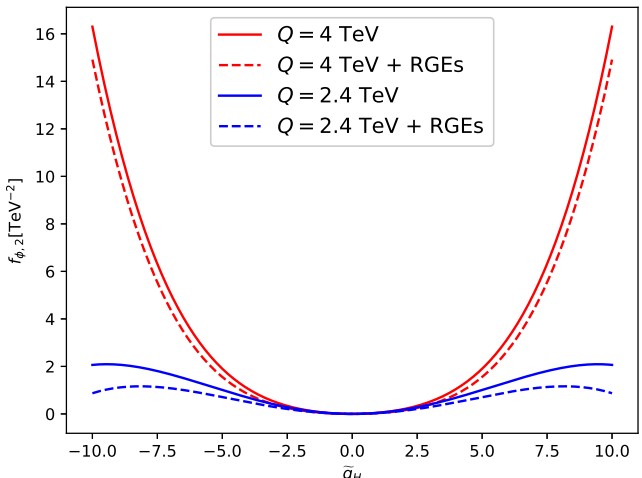

Figure 5: Wilson coefficient $f_{\phi 2}$ as a function of $\tilde{g}_H$ at different values of the matching scale $Q$ for fixed $m_V = 4\,$TeV and all other UV couplings set to zero. The dashed lines include approximate RG running.

**Vector triplet**

Moving to the triplet model defined by the Lagrangian of Eq.(10), we will not attempt to show analytic results and instead illustrate the matching scale dependence for one finite coupling $\tilde{g}_H$ and a mass term $\tilde{m}_V$ numerically. In this simplified setup, $m_V = \tilde{m}_V$. Among the various Wilson coefficients, it is instructive to consider $f_{\phi,2}$, as its dependence on the matching scale exhibits interesting features. Including both tree and loop contributions, the matching expression has the form

$$\frac{f_{\phi,2}}{\Lambda^2} \simeq \frac{1}{m_V^2}\left[ g_2^4\left(c_0 + c_1 \log\frac{m_V}{Q}\right) + \tilde{g}_H^2\left(c_2 + c_3 \log\frac{m_V}{Q}\right) + \tilde{g}_H^4\left(c_4 + c_5 \log\frac{m_V}{Q}\right)\right], \qquad (32)$$

where $c_0 = c_1/2$ emerges from 1-loop diagrams inducing the operator structure $(D_\mu W^{\mu\nu})^2$, which maps to $\mathcal{O}_{\phi,2}$ via the equations of motion. Of the additional constants, the $\tilde{g}_H^2$-coefficient is dominated by the tree-level contribution to $c_2$, while the $\tilde{g}_H^4$-coefficient is completely determined by the one-loop matching. Numerically, we find

$$c_0 = \frac{c_1}{2} = \frac{3}{128\pi^2} = 0.0024\,,$$
$$c_2 = 0.75\,, \qquad c_3 = 0.0069\,, \qquad c_4 = 0.019\,, \qquad c_5 = -0.045\,. \qquad (33)$$

In Fig. 5 we show the numerical dependence of $f_{\phi,2}$ on $\tilde{g}_H$ for different choices of $Q$. For $Q = m_V = 4\,$TeV the Wilson coefficient has a simple power dependence on $\tilde{g}_H$ driven by $c_4$. For $Q \approx 0.66\,m_V = 2.6\,$TeV the $\tilde{g}_H^4$-term cancels exactly. For $Q$ below this threshold, the coefficient in front of $\tilde{g}_H^4$ becomes negative, which flips the sign of $f_{\phi,2}$ at $\tilde{g}_H \gg 1$ and allows a solution of $f_{\phi,2} = 0$ for $\tilde{g}_H \neq 0$. For $Q \lesssim 2.4\,$TeV the solution is within the range $|\tilde{g}_H| < 4\pi$ and leads to visible effects in our global analysis.

Figure 6 shows the results of the same global analysis as in Sec. 3.1, where now we fix $m_V = 4\,$TeV. The free parameters are

$$\{\tilde{g}_H, \tilde{g}_l, \tilde{g}_M, Q\}\,, \qquad (34)$$

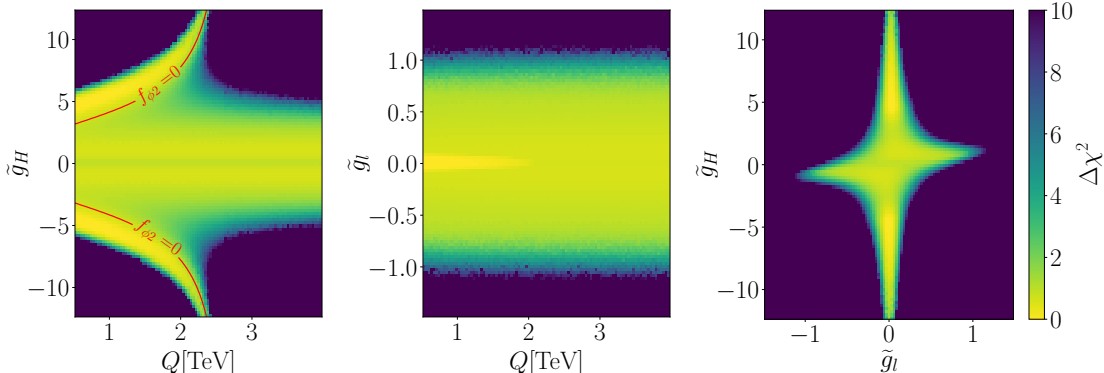

Figure 6: The impact of the variation of the matching scale $Q$ at a mass of $m_V = 4\,\text{TeV}$ for a reduced model with free $\tilde{g}_M, \tilde{g}_H, \tilde{g}_l$, expressed in the unmixed Lagrangian Eq.(15) with actual measurements.

where the matching scale is varied in the range $Q = 500\,\text{GeV} \dots 4\,\text{TeV}$. The left panel shows a central allowed region for $|\tilde{g}_H| \lesssim 4$ that is independent of $Q$. In addition, a beautiful *fleur-de-lis shape* arises in $\tilde{g}_H$ vs $Q$ for $Q < 2.4\,\text{TeV}$. It roughly follows the curves along which $f_{\phi,2} = 0$ marked in red. The Wilson coefficients $f_t, f_b, f_\tau$ have a similar behavior and vanish approximately in the same region, because they are induced by the same or similar loop contributions. As these are the operators that dominate the constraint on $\tilde{g}_H$, the fleur-de-lis feature persists in the full global fit, see Sec. 4. When we profile over $Q$ as a nuisance parameter, this correlation broadens the 1-dimensional and 2-dimensional profile likelihood in $\tilde{g}_H$ by roughly a factor 2. As shown in the second and third panels of Fig. 6, the broadening affects significantly only the constraints in the $\tilde{g}_H$ direction, while those on $\tilde{g}_l$ are essentially unchanged compared to when $Q = m_V$. Although not shown, this is also verified for $\tilde{g}_M$.

We emphasize that the tree-loop cancellations that drive this effect are only very slightly affected by the renormalization group evolution of $f_{\phi,2}$, as illustrated by the dashed lines including approximate RGE contributions in Fig. 5. They really correspond to a choice of the unphysical matching scale, which cannot be compensated by the well-defined change of renormalization scale of the low-energy SMEFT description. Adding higher orders in the loop expansion to the matching decreases the sensitivity to the matching scale. Similar effects, but with a much smaller numerical impact have been observed in Ref. [89].

## 4 SMEFT global analysis

In this section we discuss the results of the SMEFT global analysis, mapped to the parameter space of the heavy vector triplet model defined in Section 2.3 using 1-loop matching relations. We derive constraints on the UV-parameters $\{\tilde{g}_H, \tilde{g}_q, \tilde{g}_l, \tilde{g}_M, \tilde{g}_{VH}\}$ defined by the Lagrangian in Eq.(10) for fixed values of the heavy vector triplet mass. We consider two benchmark values: $m_V = 4\,\text{TeV}$, to be compared with direct resonance searches by the ATLAS Collaboration, and $m_V = 8\,\text{TeV}$ for a consistent SMEFT analysis safely below any on-shell pole.

### 4.1 Resonance searches at high invariant masses

As mentioned in Sec. 2.4, in addition to more standard Higgs measurements, the global analysis includes constraints from searches for exotic particles in the $WH$ and $WW$ channels by the ATLAS Collaboration. In particular, two of these analyses [82,83] have been newly imple-

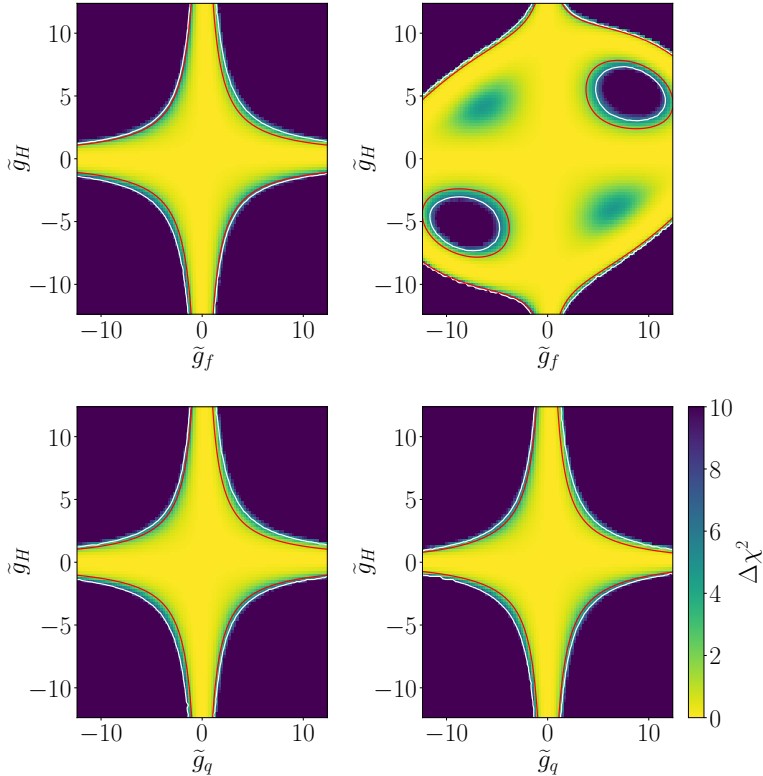

Figure 7: 2D fits of the $WH$ resonance search of Ref. [82] only. We fix $m_V = 4\,\text{TeV}$ and $\tilde{g}_M = \tilde{g}_{VH} = 0$. Left: tree-level matching. Right: Loop-level matching. Top: with $\tilde{g}_l = \tilde{g}_q = \tilde{g}_f$. Bottom: with $\tilde{g}_l = 0$. In the top (bottom) row, red contours indicate $f_W = \pm 4$ ($f_{\phi Q}^{(3)} = \pm 0.8$) with $\Lambda = 1\,\text{TeV}$ and white contours indicate $\Delta\chi^2 = 5.991$.

mented in SFITTER.

**WH search**

We consider the $m_{WH}$ invariant mass distribution measured in Ref. [82] in the $WH$ 1-tag category, and we compare it to a $WH$ signal including dimension-6 corrections. This kinematic distribution extends up to $m_{WH} = 5\,\text{TeV}$ and the strongest constraints on BSM effects stem from the region around $m_{WH} = 2 - 2.5\,\text{TeV}$, where the measurement exhibits large underfluctuations. A detailed description of the implementation of this analysis will be provided in a future work [80].

For equal values of the Wilson coefficients, the largest correction to the $m_{WH}$ spectrum is induced by the operator $\mathcal{O}_{\phi Q}^{(3)}$ [84,91–96], that contributes via corrections to the $qqV$ vertex and via a 4-point $qqVH$ interaction. The latter exhibits an enhancement at large partonic energies due to the missing $s$-channel propagator and is therefore dominant in the high-invariant-mass regime. Further significant corrections, albeit less momentum-enhanced, are induced by $\mathcal{O}_W$. All other SMEFT operators in the HISZ basis do not contribute significantly to $WH$ production in the high-energy regime.

Figure 7 shows the results from a 2D-analysis of the $m_{WH}$ distribution alone, fixing the matching scale $Q = m_V = 4\,\text{TeV}$ and considering only two $\tilde{g}$-couplings at a time. The top row in Fig. 7 shows $\tilde{g}_f \equiv \tilde{g}_q = \tilde{g}_l$ vs $\tilde{g}_H$, which matches the benchmark considered in the ATLAS analysis [82]. In this limit, the matching contribution to $f_{\phi Q}^{(3)}$ cancels exactly, both at tree and loop levels. As a consequence, the constraints are driven by $f_W$, whose matching expressions

reduce to

$$\frac{f_W}{\Lambda^2} = 4.76 \frac{\tilde{g}_H \tilde{g}_l}{m_V^2} \qquad \text{(tree)},$$

$$\frac{f_W}{\Lambda^2} \simeq \tilde{g}_l \tilde{g}_H \frac{4.71 + 0.019 \, \tilde{g}_l \tilde{g}_H - 0.023 \, \tilde{g}_l^2 - 0.057 \, \tilde{g}_H^2}{m_V^2} \qquad \text{(tree+loop)}. \qquad (35)$$

The red contours in the plots indicate $f_W/\Lambda^2 = \pm 4 \, \text{TeV}^{-2}$, which is representative of the $2\sigma$ boundaries $f_W/\Lambda^2 \in [-3.6, 4.4] \, \text{TeV}^{-2}$ found in a 1D fit to the SMEFT parameters. In a slight abuse of language, here and in the following the $\Delta\chi^2 \leq 1\,(2.3)$ and $\Delta\chi^2 \leq 3.841\,(5.991)$ regions in 1D (2D) fits are sometimes referred to as $1\sigma$ and $2\sigma$ intervals, respectively. The fact that these lines coincide to a very good approximation with the $2\sigma$ contours (indicated in white) in Fig. 7 shows that the constraint on $f_W$ is indeed the leading one. The bottom row shows $\tilde{g}_q$ vs $\tilde{g}_H$ for $\tilde{g}_l = 0$. In this case the cancellation in $f_{\phi Q}^{(3)}$ is spoiled and the constraints are dominated by this Wilson coefficient. Numerically, the matching expression is

$$\frac{f_{\phi Q}^{(3)}}{\Lambda^2} = \frac{\tilde{g}_H(\tilde{g}_l - \tilde{g}_q)}{m_V^2} \qquad \text{(tree)},$$

$$\frac{f_{\phi Q}^{(3)}}{\Lambda^2} \simeq 0.99 \frac{\tilde{g}_H(\tilde{g}_l - \tilde{g}_q)}{m_V^2} \qquad \text{(tree+loop)}, \qquad (36)$$

and the bottom panels in Fig 7 show contours for $f_{\phi Q}^{(3)}/\Lambda^2 = \pm 0.8 \, \text{TeV}^{-2}$, which is representative of the $2\sigma$ interval $f_{\phi Q}^{(3)}/\Lambda^2 \in [-0.90, 0.76] \, \text{TeV}^{-2}$ obtained in a 1D fit.

Finally, comparing the left and right panels in Fig. 7, it is worth noting that the impact of loop contributions to the matching is negligible in the case $\tilde{g}_l = 0$, but significant for $\tilde{g}_l = \tilde{g}_q$. This is a direct consequence of the form of the matching expression in the particular model considered. Loop terms only induce a very minor overall rescaling in the expression of $f_{\phi Q}^{(3)}$, Eq.(36), but they introduce a series of new terms in the expression of $f_W$, Eq.(35). Although numerically subdominant, the latter have a strong impact on the likelihood structure.

**WW search**

We consider the $m_{WW}$ distribution measured in Ref. [83] in the $WW$ 1-lepton category and ggF/DY merged, high-purity signal region, that targets neutral resonances decaying to $W^\pm W^\mp$ pairs and covers invariant masses up to $m_{WW} = 4 \, \text{TeV}$. We compare the measured distribution to a $W^\pm W^\mp$ production signal including SMEFT corrections. Again, we postpone a detailed discussion of the implementation to a later paper.

The $W^\pm W^\mp$ production process exhibits a greater complexity in the SMEFT compared to $W^\pm H$ in the high-energy limit. We find that, fixing all Wilson coefficients to the same numerical value, the largest corrections are induced by the operators $\mathcal{O}_{\phi u}, \mathcal{O}_{\phi d}, \mathcal{O}_{\phi Q}^{(1)}, \mathcal{O}_{\phi Q}^{(3)}$ at quadratic level, that exhibit a large enhancement $\propto m_{WW}^2$. The origin of this behavior can be identified as a $qq\phi\phi$ contact interaction between two quarks and two Goldstone bosons induced by these operators, that dominates at high energies due to the equivalence theorem [97]. Effects induced by $\mathcal{O}_W, \mathcal{O}_B$, and $\mathcal{O}_{WWW}$ have a weaker momentum-enhancement and are roughly two orders of magnitude smaller. Nevertheless, they were retained in the fit, as they are relevant for the global analysis in terms of both SMEFT and UV model parameters. In the former case, this measurement contributes significantly to improving the constraints on $f_W$, by roughly a factor two [80]. In the latter, it is important to stress that the matching expressions for a given UV model generally do *not* give homogeneous values for the Wilson coefficients. Therefore a

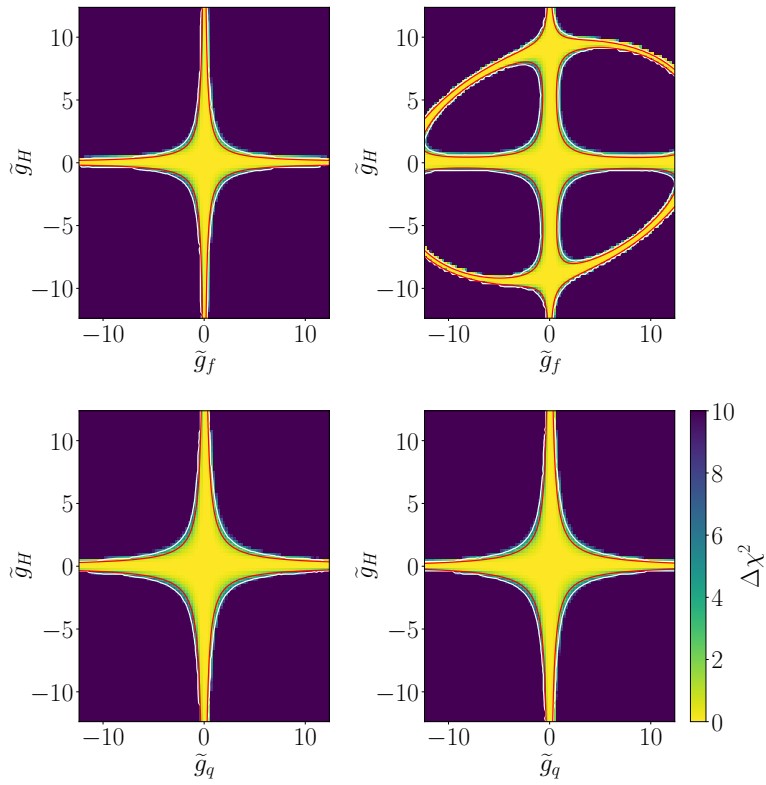

Figure 8: 2D fits of the $WW$ resonance search of Ref. [83] only. We fix $m_V = 4\,\text{TeV}$ and $\tilde{g}_M = \tilde{g}_{VH} = 0$. Left: tree-level matching. Right: Loop-level matching. Top: with $\tilde{g}_l = \tilde{g}_q = \tilde{g}_f$. Bottom: with $\tilde{g}_l = 0$. In the top (bottom) row, red contours indicate $f_W = \pm 0.7$ ($f_{\phi Q}^{(3)} = 0.2$ or $f_{\phi Q}^{(3)} = -0.3$) with $\Lambda = 1\,\text{TeV}$ and white contours indicate $\Delta\chi^2 = 5.991$.

suppression of two orders of magnitude in the SMEFT predictions can be easily compensated in the matching, and the corresponding contributions to the signal may lead to significant constraints on the UV model parameters. In fact, for the $WW$ analysis implemented here we find that the constraints projected on the $\tilde{g}_q - \tilde{g}_H$ and $\tilde{g}_f - \tilde{g}_H$ planes are entirely dominated by the contributions of $f_W$ and $f_{\phi Q}^{(3)}$, the same two operators that lead in the $WH$ case.

Figure 8 shows the results from a 2D-analysis of the $m_{WW}$ distribution alone, fixing $Q = m_V = 4\,\text{TeV}$ and considering the same benchmarks as in Fig. 7. The red curves in Fig. 8 are again given by Eq.(35) and (36), but for different values of $f_W$ and $f_{\phi Q}^{(3)}$, namely $f_W/\Lambda^2 = \pm 0.7\,\text{TeV}^{-2}$ and $f_{\phi Q}^{(3)}/\Lambda^2 = -0.27, +0.23\,\text{TeV}^{-2}$. Again, these values correspond to the $2\sigma$-boundaries identified in 1D fits.

This analysis yields stronger bounds compared to $WH$ because in this particular case the constraints are dominated by the tail of the distribution, in the region around $m_{WW} = 2.5 - 4$ TeV, which exhibits under-fluctuations. Again, the effect of introducing loop contributions to the matching expressions is only visible in the scenario dominated by $f_W$, for the same reasons as described above.

## 4.2 Global analysis results

Figure 9 shows the results of our global analysis, including the full data set described in Sec. 2.4 as well as the resonance searches discussed in Sec. 4.1, for a fixed value of the heavy vector

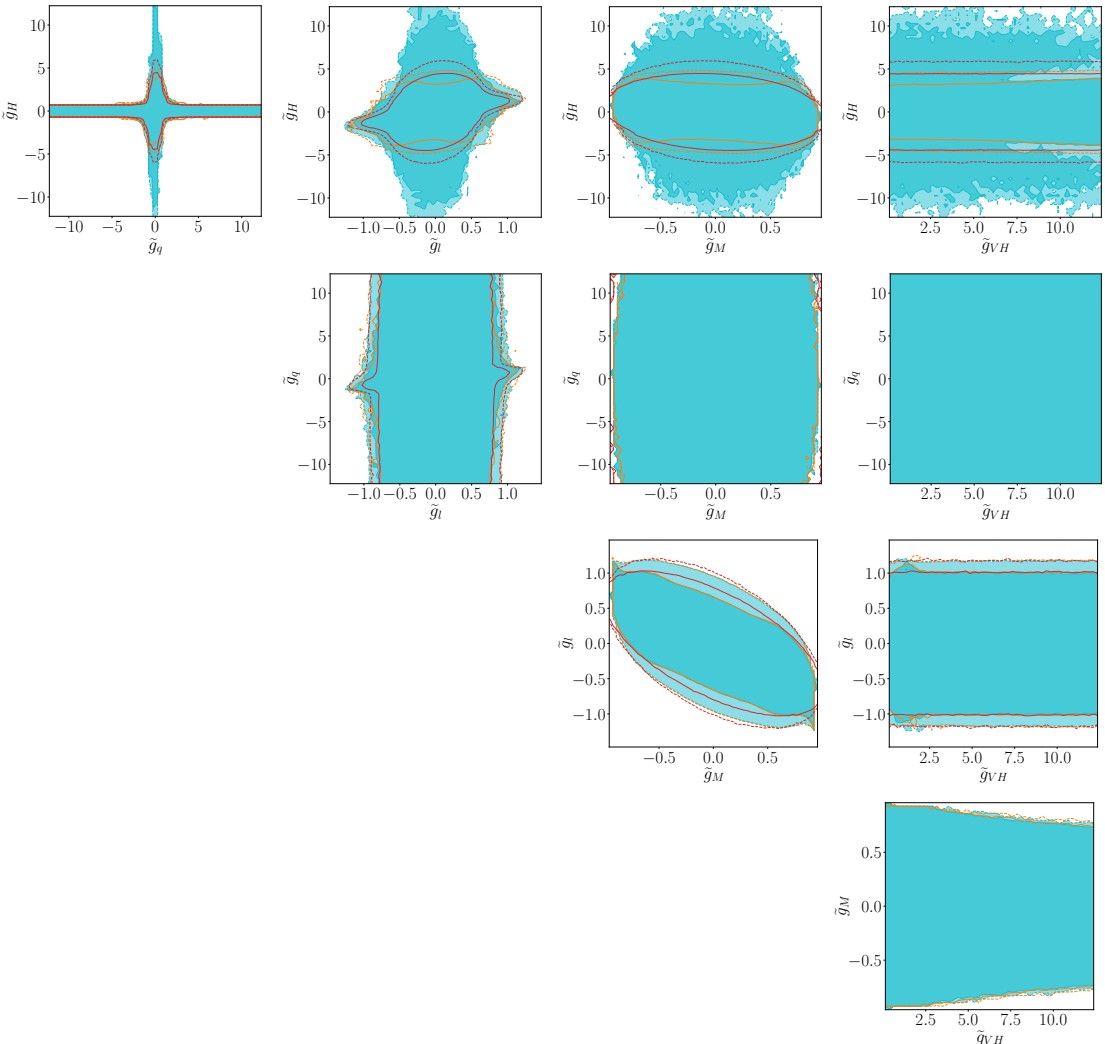

Figure 9: 5-parameter global fit of the full data set to the model parameters from Eq.(10) for fixed $m_V = 4$ TeV. Profiled $\Delta\chi^2 = 2.3$ ($\Delta\chi^2 = 5.991$) contours are shown as solid (dashed) lines. Red (orange) curves indicate the results obtained with tree (1-loop) matching onto the SMEFT and a fixed matching scale $Q = m_V$. The light blue region shows the results from 1-loop matching, profiled over $Q = 500$ GeV ... $m_V$.

triplet mass $m_V = 4$ TeV. The analysis is performed varying $\tilde{g}_M$ and $\tilde{g}_{VH}$ within the physical region $\tilde{g}_M = -1 ... 1$, $\tilde{g}_{VH} > 0$ and all other coupling parameters in the perturbative range $\tilde{g} = -4\pi ... 4\pi$.

**Fixed matching scale**

For a fixed matching scale $Q = m_V$ (red and orange lines in Fig. 9), we find that the SMEFT fit constrains significantly $\tilde{g}_l$ and $\tilde{g}_H$, while $\tilde{g}_M$, $\tilde{g}_q$, and $\tilde{g}_{VH}$ are essentially unconstrained. The striking difference between the constraints on the vector triplet couplings to leptons and to quarks is largely due to the fact that the SMEFT fit is dominated by EWPO constraints extracted at LEP, on which the leptonic interactions have a much stronger impact. We have verified that, indeed, removing EWPO constraints from the fit relaxes significantly the constraint on $\tilde{g}_l$.

The 2D projections show that $\tilde{g}_l$ is also anti-correlated to $\tilde{g}_M$. The reason is that, at tree-level, $\tilde{g}_l$ enters the matching expressions only in the combination $\tilde{g}_l + g_2 \tilde{g}_M$, where $g_2$ is the

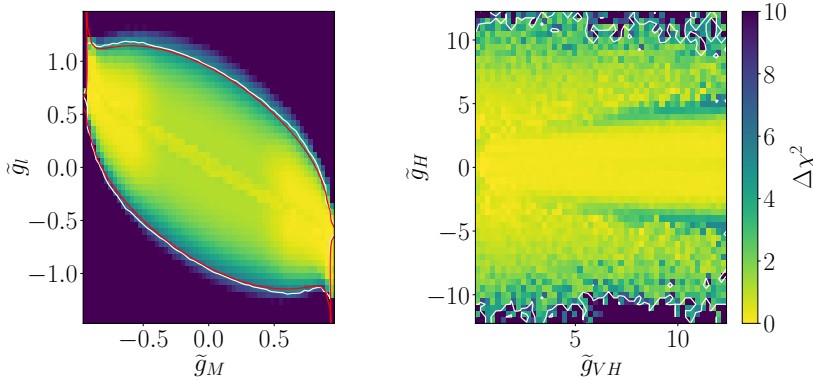

Figure 10: Heat map of the profiled $\Delta\chi^2$ distribution from the same fit as in Fig. 9, with 1-loop matching and profiling over the matching scale. The red contours indicate $f_{LLLL}/\Lambda^2 = -0.014, +0.017$ TeV$^{-2}$ and the white contours indicate $\Delta\chi^2 = 5.991$.

$SU(2)$ coupling constant. Specifically, we find that the constraints in the $\tilde{g}_M - \tilde{g}_l$ plane are dominated by the constraint on $f_{LLLL}$, whose tree-level matching expression is quadratic in the relevant combination

$$\frac{f_{LLLL}}{\Lambda^2} = -\frac{(\tilde{g}_l + g_2\tilde{g}_M)^2}{4\tilde{m}_V^2} \, . \tag{37}$$

Therefore, for most values of $\tilde{g}_M$ and $\tilde{g}_l$, the constraints are driven by the limit for negative values of this Wilson coefficient. At 1-loop, the matching expression is more complex and allows for positive values of $f_{LLLL}$ in a region close to $|\tilde{g}_M| \simeq 1$ and $|\tilde{g}_l| \simeq 1$. The right panel in Fig. 10 shows that the $2\sigma$ boundary from the 5D likelihood (in white) matches very well the contours for $f_{LLLL}/\Lambda^2 = -0.014, +0.017$ TeV$^{-2}$ (in red), corresponding to the $2\sigma$ interval derived from a 2D fit of $f_{LLLL}$ and $f_{BW}$. Here a 2D fit is necessary owing to the strong correlation between $f_{LLLL}$ and $f_{BW}$. A 1D fit would lead to an over-estimation of the constraints.

There are no major differences between tree and loop level matching when keeping the matching scale fixed $Q = m_V$. Only slight differences can be observed in the limits on $\tilde{g}_M$ and $\tilde{g}_H$. The effect on $\tilde{g}_H$ is completely washed out once the matching scale is allowed to vary, as we discuss below. Although less visible due to the different scales, an analogous anti-correlation is present in the $\tilde{g}_M - \tilde{g}_H$ plane, as $\tilde{g}_H$ also enters tree-level matching expressions exclusively in the combination $\tilde{g}_H + g_2\tilde{g}_M$. Because $\tilde{g}_H$ enters many Wilson coefficients, both at tree and loop level, in this case it is not possible to identify one particular SMEFT parameter, or combination thereof, that drives the global bounds.

The constraint on $\tilde{g}_q$, on the other hand, is driven by that on $f_{\phi Q}^{(3)}$, whose matching expression is given in Eq.(36). This is consistent with the fact that $\tilde{g}_q$ only shows a non-trivial interplay with $\tilde{g}_H$. The cross-like shape emerging in the $(\tilde{g}_q, \tilde{g}_H)$ panel results from the superposition of the hyperbola-like shape expected from the $f_{\phi Q}^3$ matching expression, and of additional constraints on $\tilde{g}_H$ that introduce extra suppressions away from the two axes. Finally, $\tilde{g}_{VH}$ does not contribute to any dimension-6 operator at tree-level, so, in this limit, the likelihood is exactly flat in the corresponding direction. At 1-loop $\tilde{g}_{VH}$ gives contributions to $f_W, f_{WW}, f_{\phi 2}, f_{t,b,\tau}$ and $f_{\phi Q}^{(3)}$. Among these, the dominant constraint stems from $f_{\phi 2}$, leading to the orange contours in the $\tilde{g}_{VH} - \tilde{g}_M$ and $\tilde{g}_{VH} - \tilde{g}_H$ planes.

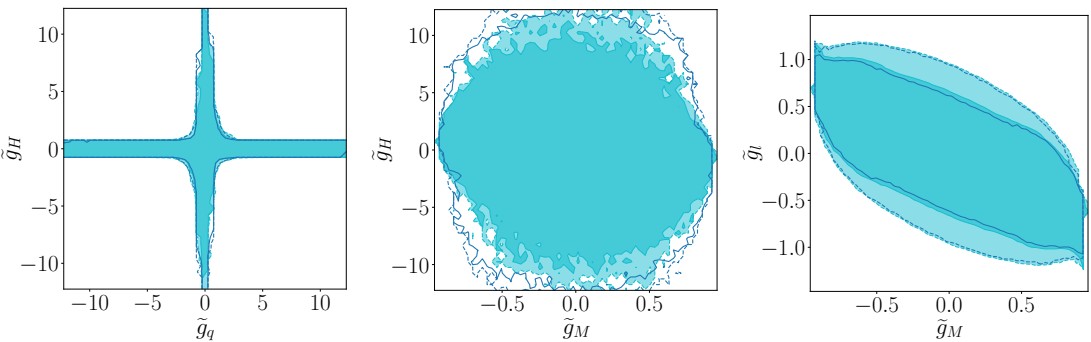

Figure 11: Impact of the high-energy kinematic distributions [81–83] on the global 5-parameter SMEFT fit for fixed $m_V = 4\,\text{TeV}$. The solid regions include the full data set (same as Fig. 9), while the dark blue lines exclude the high-energy kinematic distributions. Solid (dashed) lines mark the $\Delta\chi^2 = 2.3$ ($\Delta\chi^2 = 5.991$) contours.

**Variable matching scale**

Varying the matching scale as $Q = 500\,\text{GeV} \dots m_V = 4\,\text{TeV}$, as shown as light blue region in Fig. 9, affects the constraints on $\tilde{g}_H$, while for the other parameters the dependence is negligible. This is what we expect from the toy results in Sec. 3.2 and Fig. 6, and we have verified that extending the range to $Q \gtrsim m_V$ does not add any significant feature to the results. As for the 5-parameter fit, the main consequence of variable $Q$ is that, for $Q \lesssim 2.4\,\text{TeV}$, the matching expressions of $f_{\phi 2}$ and $f_{t,b,\tau}$ acquire a new zero. Because these operators are the dominant source of constraints on $\tilde{g}_H$, this results in a broader allowed region for this parameter, which is largest close to the $Q \simeq 2.4\,\text{TeV}$ threshold. This effect washes out the correlation between $\tilde{g}_H$ and $\tilde{g}_M$ mentioned above.

At $Q \simeq 2.4\,\text{TeV}$, the most constraining Wilson coefficient is $f_{\phi 2}$, which is responsible for the outermost region of the $2\sigma$ contours for $\tilde{g}_H$ in Fig. 9. The inner structure of the likelihood, including the $1\sigma$ contour, cannot be explained in terms of a single Wilson coefficient. It is the result of a non-trivial interplay between several effects, including $\tilde{g}_H$ entering a large number of Wilson coefficients and profiling over the matching scale.

It is also interesting to look at the finer structure of the profiled likelihood. In Fig. 10 we show $\Delta\chi^2$ for the same 2D projections as before. We can see that the best-fit points are focused in regions where $|\tilde{g}_M| > 0.5$. This effect emerges in the 5-parameter fit with 1-loop matching, irrespective of whether the matching scale is fixed or varied. It is the same effect as observed for the 3-parameter fit varying the heavy vector mass in Fig. 3, and it is due to the EWPO preferring a best-fit point away from the SM. In particular, we have checked that the observed substructures are entirely dominated by less than $1\sigma$ deviations in $A_l(SLD)$ and $m_W$. In addition, the measurements of $\sigma_h^0, R_l^0, A_{FB}^{0,l}, A_c$ reinforce the deviation through correlations. If future measurements with reduced uncertainties confirmed the present deviations from the SM, this would lead to exclusion limits with intricate patterns.

**Impact of high energy measurements**

It is well known [7] that kinematic distributions probing high invariant masses have significant impact on global fits to the SMEFT parameters. In our analysis, we confirm this behavior for the two analyses described in Sec. 4.1, which are found to constrain significantly $f_W$, $f_{\phi d}$ and $f_{WWW}$. Unfortunately, once the SMEFT is mapped onto the heavy vector triplet model, the constraining power of these measurements is diminished. This is shown in Fig. 11, where

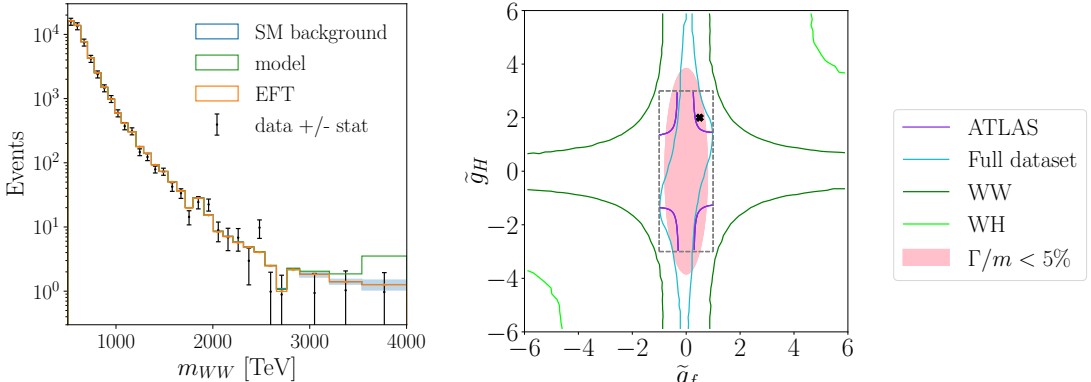

Figure 12: Left: $Z'$ prediction for $m_V = 4\,\mathrm{TeV}$, $\tilde{g}_H = 2$, $\tilde{g}_f = 0.5$ (shown by a star in the right panel) for the $WW$ search [83], compared to the SMEFT prediction. Right: SMEFT limits ($\Delta\chi^2 = 5.991$) for $m_V = 4\,\mathrm{TeV}$ and profiled over the matching scale, for the $WW$ and $WH$ distributions alone and the full dataset. We also show the 95%CL exclusion from the $WH$ resonance search [82]. The gray box marks the ATLAS search region, the narrow-width is shaded in pink.

the results of Fig. 9 are compared to those from a 5-parameter fit where the three analyses of Refs. [81–83] are removed (dark blue line). The lack of visible impact of the high-energy kinematic distributions is very much due to the specific model and the corresponding numerical behaviour of the matching formulae. As discussed above, the main constraints on the vector triplet parameter space are dominantly associated to those on $f_{LLLL}, f_{\phi 2}$ and $f_{\phi Q}^3$, which are only marginally improved by these searches.

**SMEFT vs direct searches**

A key question we would like to address in this work is whether a global SMEFT analysis can be competitive with direct searches in constraining a given UV model. Figure 12 compares the constraints in the $(\tilde{g}_f, \tilde{g}_H)$ plane obtained in the direct search of $WH$ resonances by ATLAS, Ref. [82], and from 2D SMEFT fits to different sets of observables. In particular, the light green line indicates the SMEFT constraints obtained from the same distribution as in the direct search. For all lines in this plot, the heavy triplet mass is fixed to $m_V = 4\,\mathrm{TeV}$, the maximum value accessible by the resonance search. Strictly speaking, the direct and indirect constraints extracted from the same measurement apply to complementary regions of the parameter space: the former are valid for masses $m_V \lesssim 4\,\mathrm{TeV}$ and for narrow vector triplets within the pink-shaded region of Fig 12, while the latter hold for $m_V \gg 4\,\mathrm{TeV}$ irrespective of the resonance width. Obviously, a comparison should be taken with a grain of salt.

Nevertheless, it can be instructive to examine the interplay between the signals produced by a heavy resonance and by its corresponding SMEFT approximation. The left panel of Fig. 12 shows the $m_{WW}$ resonant distribution obtained for a benchmark point at $m_V = 4\,\mathrm{TeV}$, $\tilde{g}_H = 2$, $\tilde{g}_f \equiv \tilde{g}_l = \tilde{g}_q = 0.5$, compared to the ATLAS measurement [82] (black data points) and the SMEFT signal matched to this benchmark model at dimension six. This point is indicated by a cross in Fig. 12 (right), and it is excluded at 95%CL by both the ATLAS $WH$ and $WW$ searches, but falls within the $2\sigma$-allowed region of our SMEFT global analysis. This discrepancy is obvious from the high-energy $m_{WW}$ tail, where aside from the mass peak the dimension-6 SMEFT also misses the initial rise of the distribution. Among the Wilson coefficients that contribute to $WW$ production, only $f_W/\Lambda^2 = 0.28\,\mathrm{TeV}^{-2}$ takes a value above the permille level, while $f_{\phi Q}^{(3)} = 0$ because $\tilde{g}_q = \tilde{g}_l$. This results in SMEFT signals of only a few percent

across the entire $m_{WW}$ distribution, which are always well within the uncertainties. It is worth pointing out that in such a situation the best place to look for the SMEFT signal might not just be the bins where the energy enhancement is largest, but rather those where the uncertainties are smallest.

While not surprising, these conclusions do not extend to arbitrary BSM scenarios. One characteristic of the case examined here is that the resonance is narrow. As a consequence, the effect in $m_{WW}$ is only visible close to $m_V$, where the SMEFT expansion immediately breaks down. The situation improves when we include higher-dimensional operators [22, 98]. At dimension six, the matching to our specific model suppresses all energy-enhanced SMEFT contributions to $WW$ production, so the signal is under-estimated across the e$m_{WW}$ distribution. This does not have to be the case in other BSM models. For instance, it is possible that the dimension-6 approximation over-estimates the model predictions, in which case the dimension-8 contributions need to be large and negative, and the truncated SMEFT constraints appear more stringent than those from direct searches.

Going beyond the comparison of resonance searches and SMEFT analyses for one measurement, the true power of the SMEFT approach is that it allows to combine a large number of different measurements. This will always improve the sensitivity of the SMEFT analyses and, on the other hand, it allows to derive more general conclusions, by constraining all model parameters simultaneously, as shown in Fig. 9. The light blue lines in Fig. 12 show the constraints from a 2-parameter SMEFT fit to the entire dataset employed in this work. Consistent with the discussions above, these limits are dominated by EWPO, for which the SMEFT expansion is valid. In particular, the constraint on $\tilde{g}_f$ is dominated by the leptonic component $\tilde{g}_l$, which in turn is mostly associated to the $f_{LLLL}$ Wilson coefficient. Comparing these limits to those from the ATLAS $WH$-search, we find that the latter are slightly stronger for $|\tilde{g}_H| \gtrsim 1$ (with the caveat that they are only valid in the narrow width regime), while the former dominate for $|\tilde{g}_H| \lesssim 1$. Here, the $WH$ search has an unconstrained direction along the $\tilde{g}_H = 0$ axis, that is broken by the EWPO in the SMEFT fit [25].

**Heavy vector results**

One of the main motivations for the SMEFT formalism is that it allows us to derive constraints on new particles with masses beyond the reach of direct searches. In this spirit, we can extend our SMEFT constraints on the $\tilde{g}$ parameters for a heavy triplet mass to $m_V = 8$ TeV. Now, the dimension-6 SMEFT approximation is valid all over the kinematic measurements discussed above. The corresponding results in Fig. 13 can be directly compared to those in Fig. 9 for $m_V = 4$ TeV. As expected, all the bounds on the model parameters are weaker for heavier values of $m_V$ (see also Fig. 3). However, a notable feature is that the limits do not simply scale with a factor proportional to $m_V$, as one would naively expect from the SMEFT analysis at dimension six. The reason is that the matching expressions that relate the Wilson coefficients to the model parameters are generally non-trivial and do not scale universally with $(\tilde{g}_i/m_V)$, as can be seen for instance in Eq.(32). Moreover, as we are considering a BSM state that is not a singlet under $SU(2)$, the EW gauge coupling $g_2$ contributes to the matching independently of the $\tilde{g}$ parameters. The result is that the degeneracy between $\tilde{g}_i$ and $m_V$ is largely broken in the matching, leading to a complex likelihood structure that changes significantly with $m_V$.

## 5 Conclusions

We have presented a global analysis of a Standard Model extension with a gauge-triplet vector resonance in terms of the dimension-6 SMEFT Lagrangian. We have performed a global SFIT-TER analysis including electroweak precision observables, Higgs and di-boson measurements

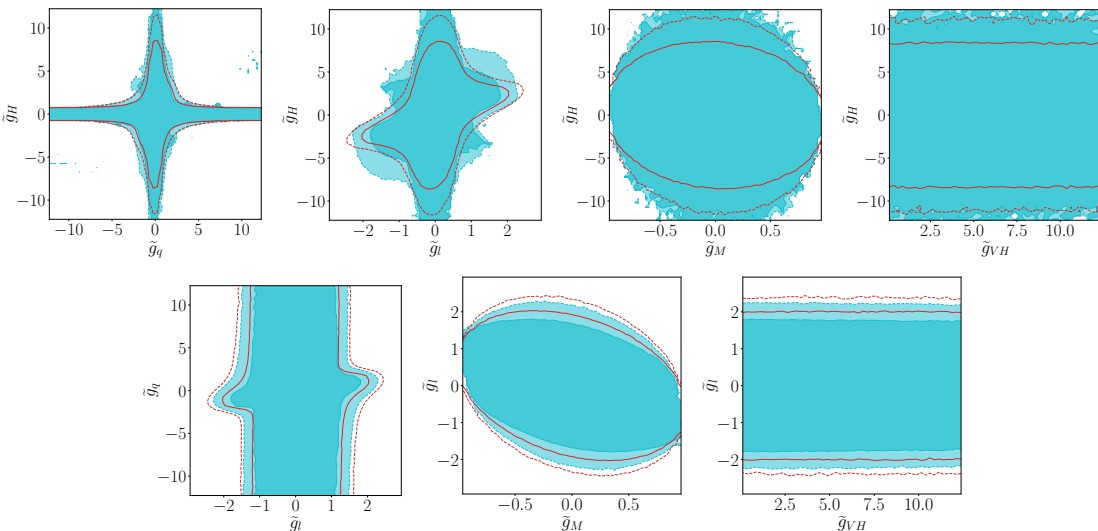

Figure 13: 5-parameter fit to the full data set for the model parameters in Eq.(10) for fixed $m_V = 8$ TeV. Each panel shows profiled $\Delta\chi^2 = 2.3$ (solid) and $\Delta\chi^2 = 5.991$ (dashed) contours. Red curves correspond to tree-level matching, the light blue region to 1-loop matching, profiled over three $\tilde{g}$ parameters plus the matching scale $Q = 500$ GeV ... $m_V$. The panels for $\tilde{g}_M - \tilde{g}_q$ $\tilde{g}_{VH} - \tilde{g}_q$ and $\tilde{g}_{VH} - \tilde{g}_M$ are not shown as they are unconstrained in the explored ranges.

as well as resonance searches at the LHC, and have compared our results with limits obtained from direct searches. To relate the full model and the SMEFT we have employed one-loop matching with a focus on the theory uncertainties from the choice of the matching scale.

First, we have shown that the theory uncertainty due to the choice of the matching scale can have a large effect on the global analysis. In particular, the bounds on the coupling of the new vector to the SM-Higgs are significantly weakened once we profile over a variable matching scale, illustrating how all theory uncertainties need to be taken into account at least once we translate SMEFT results back into models.

Comparing the SMEFT results with direct searches reveals an intriguing complementarity. Direct and SMEFT searches are reliable in different parameter regions; while direct searches are sensitive to narrow resonances with kinematically accessible masses, SMEFT searches apply to energies sufficiently below the resonance mass. The SMEFT analysis can be sensitive to the onset of the resonance, but a reliable description of this region requires a tower of higher-dimensional operators. Specifically for the vector-triplet model, the SMEFT model for the high-energy tail of kinematic distributions turned out less sensitive than the resonance search, and therefore provided conservative constraints. On the other hand, the SMEFT analysis can probe vector masses beyond the reach of resonance searches. Here, we found that the one-loop matching dampens the sensitivity decrease of the SMEFT analysis compared to the naively expected scaling.

While SMEFT analyses cannot replace model-specific searches for new physics, they add valuable constraints from a large variety of measurements and are sensitive to new physics scales beyond the reach of resonance searches. Only this complementarity of direct and indirect searches allows us to make best use of current and future LHC data.

# Acknowledgments

We would like to thank Anke Biekötter, Anja Butter, Tyler Corbett and David Lopez-Val for extensive support with SFITTER and Inês Ochoa for her help with the *VH* ATLAS analysis and for updating the corresponding HEPData repository. MK and BS are grateful to Alexander Voigt for discussions on one-loop matching. We acknowledge support by the state of Baden-Württemberg through bwHPC and the German Research Foundation (DFG) through grant no INST 39/963-1 FUGG (bwForCluster NEMO). BS was partially supported by the German Research Foundation(DFG) by the grant STU 615/2-1. The research of everyone is supported by the Deutsche Forschungsgemeinschaft (DFG, German Research Foundation) under grant 396021762 – TRR 257 *Particle Physics Phenomenology after the Higgs Discovery*.

# A Appendix

## A.1 Operator basis

We consider the dimension-6 SMEFT Lagrangian

$$\mathcal{L}_{\text{SMEFT}} \supset -\frac{\alpha_s}{8\pi}\frac{f_{GG}}{\Lambda^2}\mathcal{O}_{GG} + \frac{f_{WW}}{\Lambda^2}\mathcal{O}_{WW} + \frac{f_{BB}}{\Lambda^2}\mathcal{O}_{BB} + \frac{f_{BW}}{\Lambda^2}\mathcal{O}_{BW} + \frac{f_W}{\Lambda^2}\mathcal{O}_W + \frac{f_B}{\Lambda^2}\mathcal{O}_B$$
$$+ \frac{f_{WWW}}{\Lambda^2}\mathcal{O}_{WWW} + \frac{f_{\phi1}}{\Lambda^2}\mathcal{O}_{\phi1} + \frac{f_{\phi2}}{\Lambda^2}\mathcal{O}_{\phi2} + \frac{f_\tau m_\tau}{v\Lambda^2}\mathcal{O}_\tau + \frac{f_b m_b}{v\Lambda^2}\mathcal{O}_b + \frac{f_t m_t}{v\Lambda^2}\mathcal{O}_t$$
$$+ \frac{f_{LLLL}}{\Lambda^2}\mathcal{O}_{LLLL} + \frac{f_{\phi e}}{\Lambda^2}\mathcal{O}_{\phi e} + \frac{f_{\phi d}}{\Lambda^2}\mathcal{O}_{\phi d} + \frac{f_{\phi u}}{\Lambda^2}\mathcal{O}_{\phi u} + \frac{f_{\phi Q}^{(1)}}{\Lambda^2}\mathcal{O}_{\phi Q}^{(1)} + \frac{f_{\phi Q}^{(3)}}{\Lambda^2}\mathcal{O}_{\phi Q}^{(3)}, \quad (\text{A.1})$$

where the Wilson coefficients are denoted by $f_i$. We use the dimension-6 operator basis of Ref. [7], which is based on the HISZ set [78] and defined in Tab. 1. We adopt the '+' convention for the covariant derivatives, e.g. $D_\mu\phi = (\partial_\mu + ig'B_\mu/2 + igt^A W_\mu^A)\phi$, where $t^A = \sigma^A/2$ are the $SU(2)$ generators and $\sigma^A$ the Pauli matrices. We have also defined

Table 1: Basis of dimension-6 SMEFT operators adopted in our global analysis. Flavor indices are denoted by $i, j$ and are implicitly contracted when repeated.

| | |
|---|---|
| $\mathcal{O}_{GG} = \phi^\dagger\phi G_{\mu\nu}^a G^{a\mu\nu}$ | $\mathcal{O}_{BW} = \phi^\dagger\hat{B}_{\mu\nu}\hat{W}^{\mu\nu}\phi$ |
| $\mathcal{O}_{BB} = \phi^\dagger\hat{B}_{\mu\nu}\hat{B}^{\mu\nu}\phi$ | $\mathcal{O}_{WW} = \phi^\dagger\hat{W}_{\mu\nu}\hat{W}^{\mu\nu}\phi$ |
| $\mathcal{O}_B = (D_\mu\phi)^\dagger\hat{B}^{\mu\nu}(D_\nu\phi)$ | $\mathcal{O}_W = (D_\mu\phi)^\dagger\hat{W}^{\mu\nu}(D_\nu\phi)$ |
| $\mathcal{O}_{WWW} = \text{Tr}(\hat{W}_{\mu\nu}\hat{W}^{\nu\rho}\hat{W}_\rho^{\ \mu})$ | |
| $\mathcal{O}_{\phi1} = (D_\mu\phi)^\dagger\phi\phi^\dagger(D^\mu\phi)$ | $\mathcal{O}_{\phi2} = \frac{1}{2}\partial^\mu(\phi^\dagger\phi)\partial_\mu(\phi^\dagger\phi)$ |
| $\mathcal{O}_b = (\phi^\dagger\phi)\bar{q}_3\phi d_3$ | $\mathcal{O}_\tau = (\phi^\dagger\phi)\bar{l}_3\phi e_3$ |
| $\mathcal{O}_t = (\phi^\dagger\phi)\bar{q}_3\tilde{\phi}u_3$ | |
| $\mathcal{O}_{LLLL} = (\bar{l}_1\gamma_\mu l_2)(\bar{l}_2\gamma^\mu l_1)$ | $\mathcal{O}_{\phi e} = (\phi^\dagger i\overleftrightarrow{D}_\mu\phi)(\bar{e}_i\gamma^\mu e_j)\delta^{ij}$ |
| $\mathcal{O}_{\phi d} = (\phi^\dagger i\overleftrightarrow{D}_\mu\phi)(\bar{d}_i\gamma^\mu d_j)\delta^{ij}$ | $\mathcal{O}_{\phi u} = (\phi^\dagger i\overleftrightarrow{D}_\mu\phi)(\bar{u}_i\gamma^\mu u_j)\delta^{ij}$ |
| $\mathcal{O}_{\phi Q}^{(1)} = (\phi^\dagger i\overleftrightarrow{D}_\mu\phi)(\bar{q}_i\gamma^\mu q_j)\delta^{ij}$ | $\mathcal{O}_{\phi Q}^{(3)} = (\phi^\dagger i\overleftrightarrow{D}_\mu^A\phi)(\bar{q}_i\gamma^\mu t^A q_j)\delta^{ij}$ |

$(\phi^\dagger i \overleftrightarrow{D}_\mu \phi) = i\phi^\dagger(D_\mu \phi) - i(D_\mu \phi^\dagger)\phi$ , $(\phi^\dagger i \overleftrightarrow{D}_\mu^I \phi) = i\phi^\dagger t^A(D_\mu \phi) - i(D_\mu \phi^\dagger)t^A \phi$ and the dual Higgs field $\widetilde{\phi} = i\sigma^2 \phi^\star$. The field strengths are normalized as $\hat{B}_{\mu\nu} = ig'B_{\mu\nu}/2$ and $\hat{W}_{\mu\nu} = ig t^A W_{\mu\nu}^A$. Finally, the operators $\mathcal{O}_{\phi Q}^{(1),(3)}, \mathcal{O}_{\phi u}, \mathcal{O}_{\phi d}, \mathcal{O}_{\phi e}$ are defined in a $U(3)^5$-invariant flavor structure, while for $\mathcal{O}_{LLLL}$ we only retain the (1221) contraction, that is relevant for the definition of the Fermi constant, and for $\mathcal{O}_b, \mathcal{O}_t, \mathcal{O}_\tau$ we only consider the 3rd fermion generation. The latter choice is justified considering that, in a $U(3)^5$-symmetric scenario, these operators are weighted by a Yukawa coupling insertion, that acts as a suppression for the first two families.

The matching to the UV models described in Sec. 2.1 is automated for the Warsaw basis of SMEFT operators [99], in the general flavor case. The results obtained are provided on github at [79] and we give explicit expressions for the tree-level matching in Appendix A.2. In order to interface them to SFITTER, the matching results are mapped onto the basis of Tab. 1. In the following we denote the operators in the Warsaw basis, defined as in Ref. [99], by $Q_k$ and the associated Wilson coefficients by $C_k$, such that the SMEFT Lagrangian in this basis has the form

$$\mathcal{L}_{\text{Warsaw}} \supset \frac{1}{\Lambda^2} \sum_k \sum_{ij} C_{k,ij} Q_{k,ij}, \tag{A.2}$$

where $k$ runs over the operators labels and $i, j$ are flavor indices, that are present for fermionic operators. The relations between the two operator bases are

$$\mathcal{O}_{GG} = Q_{\phi G}, \qquad \mathcal{O}_{WWW} = \frac{g^3}{4} Q_W,$$

$$\mathcal{O}_{BB} = -\frac{g'^2}{4} Q_{\phi B}, \qquad \mathcal{O}_{WW} = -\frac{g^2}{4} Q_{\phi W}, \qquad \mathcal{O}_{BW} = -\frac{g g'}{4} Q_{\phi WB},$$

$$\mathcal{O}_{\phi 1} = Q_{\phi D}, \qquad \mathcal{O}_{\phi 2} = -\frac{1}{2} Q_{\phi\Box}, \qquad \mathcal{O}_\phi = Q_\phi,$$

$$\mathcal{O}_\tau = Q_{e\phi,33}, \qquad \mathcal{O}_t = Q_{u\phi,33}, \qquad \mathcal{O}_b = Q_{d\phi,33},$$

$$\mathcal{O}_{\phi e} = Q_{\phi e,ij}\,\delta^{ij}, \qquad \mathcal{O}_{\phi u} = Q_{\phi u,ij}\,\delta^{ij}, \qquad \mathcal{O}_{\phi d} = Q_{\phi d,ij}\,\delta^{ij},$$

$$\mathcal{O}_{\phi Q}^{(1)} = Q_{\phi q,ij}^{(1)}\,\delta^{ij}, \qquad \mathcal{O}_{\phi Q}^{(3)} = \frac{1}{4} Q_{\phi q,ij}^{(3)}\,\delta^{ij}, \qquad \mathcal{O}_{LLLL} = Q_{ll,1221}, \tag{A.3}$$

and

$$\mathcal{O}_W = \frac{g^2}{8} Q_{\phi W} + \frac{g' g}{8} Q_{\phi WB} - \frac{3g^2}{8} Q_{\phi\Box} + \frac{g^2 m_h^2}{4}(\phi^\dagger \phi)^2 - \frac{g^2 \lambda}{2} Q_\phi$$
$$- \frac{g^2}{4}\left[ (Y_e)_{ij} Q_{e\phi,ij} + (Y_u)_{ij} Q_{u\phi,ij} + (Y_d)_{ij} Q_{d\phi,ij} + \text{h.c.} \right] - \frac{g^2}{8}\left( Q_{\phi q,ij}^{(3)} + Q_{\phi l,ij}^{(3)} \right)\delta^{ij},$$

$$\mathcal{O}_B = \frac{g'^2}{8} Q_{\phi B} + \frac{g g'}{8} Q_{\phi WB} - \frac{g'^2}{2} Q_{\phi D} - \frac{g'^2}{8} Q_{\phi\Box}$$
$$- \frac{g'^2}{4}\left( \frac{1}{6} Q_{\phi q,ij}^{(1)} - \frac{1}{2} Q_{\phi l,ij}^{(1)} + \frac{2}{3} Q_{\phi u,ij} - \frac{1}{3} Q_{\phi d,ij} - Q_{\phi e,ij} \right)\delta^{ij}, \tag{A.4}$$

where all repeated flavor indices are implicitly summed over, and $\lambda$ is the quartic coupling in the Higgs potential, normalised such that

$$V(\phi) = -\frac{m_h^2}{2}\phi^\dagger \phi + \frac{\lambda}{2}(\phi^\dagger \phi)^2. \tag{A.5}$$

As the vector triplet model we are interested in is defined in a flavor-symmetric limit, after the matching procedure the Wilson coefficients of the Warsaw basis operators $Q_{\phi e,\phi u,\phi d}$ and

$Q_{\phi l, \phi q}^{(1),(3)}$ will have the form

$$C_{\phi\psi,ij} = \bar{C}_{\phi\psi}\,\delta_{ij}\,, \tag{A.6}$$

while

$$C_{ll,ijkl} = \bar{C}_{ll}\delta_{ij}\delta_{kl} + \bar{C}'_{ll}\delta_{il}\delta_{kj}\,. \tag{A.7}$$

Using this notation, the mapping in terms of Wilson coefficients is

$$
\begin{aligned}
f_B &= \frac{8}{g'^2}\bar{C}_{\phi l}^{(1)}\,, & -\frac{\alpha_s}{8\pi}f_{GG} &= C_{\phi G}\,, \\
f_W &= -\frac{8}{g^2}\bar{C}_{\phi l}^{(3)}\,, & f_{WWW} &= \frac{4}{g^3}C_W\,, \\
f_{BB} &= -\frac{4}{g'^2}\left[C_{\phi B} - \bar{C}_{\phi l}^{(1)}\right]\,, & f_{\phi 1} &= C_{\phi D} + 4\bar{C}_{\phi l}^{(1)}\,, \\
f_{WW} &= -\frac{4}{g^2}\left[C_{\phi W} + \bar{C}_{\phi l}^{(3)}\right]\,, & f_{\phi 2} &= -2C_{\phi\Box} - 2\bar{C}_{\phi l}^{(1)} + 6\bar{C}_{\phi l}^{(3)}\,, \\
f_{BW} &= 4\left[-\frac{C_{\phi WB}}{g g'} - \frac{\bar{C}_{\phi l}^{(3)}}{g^2} + \frac{\bar{C}_{\phi l}^{(1)}}{g'^2}\right]\,, & f_\phi &= C_\phi - 4\lambda\bar{C}_{\phi l}^{(3)}\,, \tag{A.8}
\end{aligned}
$$

and for the fermionic ones

$$
\begin{aligned}
\frac{m_\tau}{v}f_\tau &= C_{e\phi,33} - 2(Y_e)_{33}\bar{C}_{\phi l}^{(3)}\,, & f_{\phi e} &= \bar{C}_{\phi e} - 2\bar{C}_{\phi l}^{(1)}\,, \\
\frac{m_t}{v}f_t &= C_{u\phi,33} - 2(Y_u)_{33}\bar{C}_{\phi l}^{(3)}\,, & f_{\phi u} &= \bar{C}_{\phi u} + \frac{4}{3}\bar{C}_{\phi l}^{(1)}\,, \\
\frac{m_b}{v}f_b &= C_{d\phi,33} - 2(Y_d)_{33}\bar{C}_{\phi l}^{(3)}\,, & f_{\phi d} &= \bar{C}_{\phi d} - \frac{2}{3}\bar{C}_{\phi l}^{(1)}\,, \\
f_{\phi Q}^{(1)} &= \bar{C}_{\phi q}^{(1)} + \frac{1}{3}\bar{C}_{\phi l}^{(1)}\,, & f_{\phi Q}^{(3)} &= 4\left[\bar{C}_{\phi q}^{(3)} - \bar{C}_{\phi l}^{(3)}\right]\,, \\
f_{LLLL} &= \bar{C}'_{ll}\,. & & \tag{A.9}
\end{aligned}
$$

In addition, the Higgs quartic coupling gets redefined as

$$\lambda_{\text{HISZ}} = \lambda_{\text{Warsaw}} + \frac{4m_h^2}{\Lambda^2}\bar{C}_{\phi l}^{(3)}\,. \tag{A.10}$$

This translates into corrections to the cubic and quartic Higgs self-couplings, which do not contribute to any of the observables in our fit.

## A.2 Matching expressions at tree-level

Matching the heavy vector triplet model defined in Section 2.3 at tree level onto the Warsaw basis, we obtain

$$C_{\phi\square} = -\frac{3}{8}\frac{(\tilde{g}_H + g_2\tilde{g}_M)^2}{\tilde{m}_V^2},$$

$$C_{\phi l,ij}^{(3)} = \bar{C}_{\phi l}^{(3)}\delta_{ij} = -\frac{1}{4}\frac{(\tilde{g}_l + g_2\tilde{g}_M)(\tilde{g}_H + g_2\tilde{g}_M)}{\tilde{m}_V^2}\delta_{ij},$$

$$C_{\phi Q,ij}^{(3)} = \bar{C}_{\phi q}^{(3)}\delta_{ij} = -\frac{1}{4}\frac{(\tilde{g}_q + g_2\tilde{g}_M)(\tilde{g}_H + g_2\tilde{g}_M)}{\tilde{m}_V^2}\delta_{ij},$$

$$C_{ll,ijkl} = \bar{C}_{ll}\delta_{ij}\delta_{kl} + \bar{C}_{ll}'\delta_{il}\delta_{kj} = \frac{1}{8}\frac{(\tilde{g}_l + g_2\tilde{g}_M)^2}{\tilde{m}_V^2}\left(\delta_{ij}\delta_{kl} - 2\delta_{il}\delta_{kj}\right),$$

$$C_{f\phi,ij} = -\frac{(Y_f)_{ij}}{4}\frac{(\tilde{g}_H + g_2\tilde{g}_M)^2}{\tilde{m}_V^2} \qquad (f = e, u, d). \qquad \text{(A.11)}$$

These results were also derived e.g. in Refs. [14, 24, 27, 100]. The full expressions for the 1-loop matching are derived here for the first time and are provided at Ref. [79].

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
