# Peer review of "From Models to SMEFT and Back?"

_SciPost Physics, doi:SciPost Phys. 12, 036 (2022)_

## Round 1 · Referee Report · Anonymous (Referee 1) · 2021-10-13

Report
This papers deals with the Standard Model Effective Field Theory (SMEFT). The authors attempt to answer to a series of questions: How far can we go with the SMEFT in the interpretation of its results? How precise are its predictions? With the increase of the collider energies, is SMEFT still useful? To answer to these questions, they compares the effective Lagrangian of SMEFT with some ultraviolet (UV) completions: introducing to the SM spectrum either a scalar singlet or a vector $SU(2)_L$ triplet, and performing the matching at 1-loop order.
Besides, the technical details of the matching, that look correct, I appreciated the discussion on the uncertainties on the theoretical predictions due to the free scales present in the theory: the renormalisation scale and the matching scale. In particular, the choice of the matching scale have relevant impact in the bounds that can be extracted on the new vector field.
Coming to the comparison between the purely SMEFT global fit and the full model analysis, the results are the expected ones: close to the mass of the new particles, direct searches are the most appropriate and accurate analyses; at higher energies, instead, the SMEFT analysis is sensitive to new physics beyond the reach of direct searches.
The paper is clearly written, rich in details in the analytical expressions and in the numerical results. Some of the results were expected, although there are also some new original parts. In my opinion the paper can be accepted for publication as it represents a good example of how to deal with SMEFT and UV models without forgetting relevant details, such as the dependence on the matching scale.
Besides, the technical details of the matching, that look correct, I appreciated the discussion on the uncertainties on the theoretical predictions due to the free scales present in the theory: the renormalisation scale and the matching scale. In particular, the choice of the matching scale have relevant impact in the bounds that can be extracted on the new vector field.
Coming to the comparison between the purely SMEFT global fit and the full model analysis, the results are the expected ones: close to the mass of the new particles, direct searches are the most appropriate and accurate analyses; at higher energies, instead, the SMEFT analysis is sensitive to new physics beyond the reach of direct searches.
The paper is clearly written, rich in details in the analytical expressions and in the numerical results. Some of the results were expected, although there are also some new original parts. In my opinion the paper can be accepted for publication as it represents a good example of how to deal with SMEFT and UV models without forgetting relevant details, such as the dependence on the matching scale.

---

## Round 1 · Referee Report · jacky kumar (Referee 2) · 2021-10-14

Strengths
1-The analyses presented in this paper are comprehensive, complete, important, and timely.
Report
In this paper, the authors have performed a rigorous global analysis of the triplet extension of the Standard Model. They have used the SMEFT framework for confronting this model to the Electroweak precision, Higgs, and Diboson measurements at the LHC. For this purpose, the 1-loop matching to the SMEFT, which is performed by employing the functional techniques, is presented for the first time in this paper. I would recommend publishing this paper in SciPost Physics.

---

## Editorial Decision

published